# TP-Spikformer: Token Pruned Spiking Transformer

**Wenjie Wei**[1], **Xiaolong Zhou**[1], **Malu Zhang**[1,2]\*, **Ammar Belatreche**[3], **Qian Sun**[1],
**Yimeng Shan**[1], **Dehao Zhang**[1], **Zijian Zhou**[1], **Zeyu Ma**[1], **Yang Yang**[1], **Haizhou Li**[2,4]
[1]University of Electronic Science and Technology of China, [2]Shenzhen Loop Area Institute,
[3]Northumbria University, [4]The Chinese University of Hong Kong, Shenzhen (CUHK-Shenzhen)

## ABSTRACT

Spiking neural networks (SNNs) offer an energy-efficient alternative to traditional neural networks due to their event-driven computing paradigm. However, recent advancements in spiking transformers have focused on improving accuracy with large-scale architectures, which require significant computational resources and limit deployment on resource-constrained devices. In this paper, we propose a simple yet effective token pruning method for spiking transformers, termed TP-Spikformer, that reduces storage and computational overhead while maintaining competitive performance. Specifically, we first introduce a heuristic spatiotemporal information-retaining criterion that comprehensively evaluates tokens' importance, assigning higher scores to informative tokens for retention and lower scores to uninformative ones for pruning. Based on this criterion, we propose an information-retaining token pruning framework that employs a block-level early stopping strategy for uninformative tokens, instead of removing them outright. This also helps preserve more information during token pruning. We demonstrate the effectiveness, efficiency and scalability of TP-Spikformer through extensive experiments across diverse architectures, including Spikformer, QKFormer and Spike-driven Transformer V1 and V3, and a range of tasks such as image classification, object detection, semantic segmentation and event-based object tracking. Particularly, TP-Spikformer performs well in a training-free manner. These results reveal its potential as an efficient and practical solution for deploying SNNs in real-world applications with limited computational resources.

## 1 INTRODUCTION

Spiking Neural Networks (SNNs) have emerged as a promising energy-efficient solution for next-generation machine intelligence Gerstner & Kistler (2002); Izhikevich (2003). In SNNs, the discrete binary spike serves as the fundamental information carrier and is conveyed event-drivenly. This unique computing paradigm allows only a subset of neurons to be activated and engage in synaptic accumulation operations, achieving significant computational efficiency Pfeiffer & Pfeil (2018); Roy et al. (2019); Li et al. (2024). In addition, the sparse event-driven nature of SNNs has spurred the development of neuromorphic hardware, such as TrueNorth Akopyan et al. (2015) and Loihi Davies et al. (2018), further harnessing their potential for energy efficiency. Despite the significant efficiency advantages, the limited performance of SNNs presents challenges to their widespread applications.

Building on the success of Transformer models across various fields Devlin et al. (2019); Dosovitskiy et al. (2020), researchers have integrated them with SNNs, such as Spikformer Zhou et al., QKFormerZhang et al. (2024), Spike-driven transformer(SDT)-V1 Yao et al. (2023a), V2 Yao et al. and V3 Yao et al. (2025), leading to significant performance improvements on large and complex benchmarks. However, these gains come at the expense of a large number of model parameters and high computational complexity. For example, the recently introduced SDT-V3 Yao et al. (2025) achieves 86.2% accuracy on ImageNet. Yet, this model contains 173 million parameters, requires 1384MB of memory, and performs 28.4 billion synaptic operations per second during inference.

---

\*Corresponding author: maluzhang@uestc.edu.cn

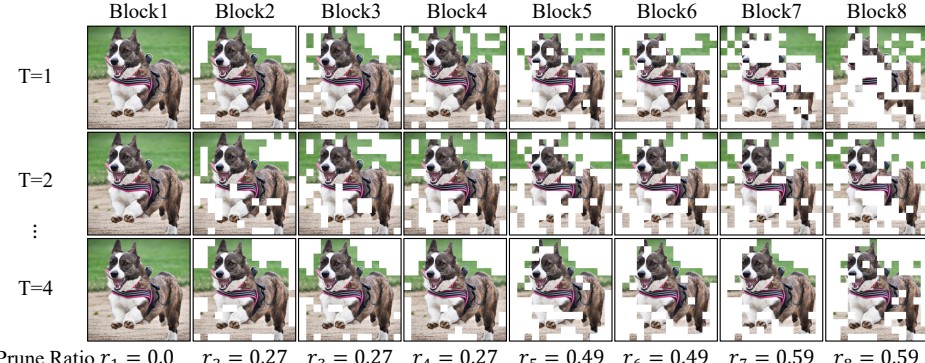

Figure 1: Visualization of token pruning across time step and block with our method. Experiments are conducted on SDT-V1-8-512, and white areas are pruned tokens.

These present significant challenges for deploying Transformer-based SNNs on resource-limited scenarios Qiu et al. (2025); Zhan et al. (2025).

Researchers have made significant efforts to compress large-scale spiking transformers, including techniques of quantization Cao et al. (2025); Wei et al. (2024a), network architecture search Wang et al. (2024b); Che et al. (2024); Zhang et al. (2025), and pruning Zhuge et al. (2024); Zhou et al. (2024c). Among these, token pruning dynamically reduces the number of tokens processed in each block during inference, enhancing both storage and computational efficiency. The underlying principle behind it is that, in visual tasks, the final prediction typically relies on only a subset of the tokens. This allows us to selectively remove certain tokens, accelerating inference while maintaining competitive accuracy Rao et al. (2021); Yin et al. (2022). Therefore, token pruning poses a promising solution for the efficient deployment of spiking transformers, particularly in edge scenarios. However, current token pruning methods in SNNs suffer from two major limitations Zhuge et al. (2024); Liu et al. (2024); Kang et al. (2024). First, most approaches modify the original structure when applied to spiking transformers, such as introducing tokens, adding trainable modules, or altering network connections. Second, these methods typically require retraining the model, resulting in large training costs. These issues raise the application costs and reduce their generalizability.

In this paper, we propose a simple yet effective token pruning approach for spiking transformer (TP-Spikformer), aiming to compress its storage and accelerate its computation while maintaining competitive performance. We first propose a heuristic spatiotemporal information-retaining token pruning criterion (IRToP), where informative tokens are assigned higher scores to retain and uninformative tokens are given lower scores to prune. Based on this criterion, we design an information-retaining token pruning architecture (IR-Arc), which achieves compression and acceleration by applying a block-level early stopping strategy for uninformative tokens, rather than direct dropping. This also helps retain more information during token pruning. The token pruning results of TP-Spikformer are depicted in Figure 1, and our main contributions are summarized as follows:

- We propose a heuristic spatiotemporal information-retaining criterion for token pruning, termed IRToP. Spatially, IRToP recognizes tokens that differ significantly from neighbors as more distinctive. Temporally, IRToP identifies tokens with greater variation across adjacent time steps as carriers of richer temporal information. By integrating both aspects, IRToP effectively identifies informative tokens and assigns them higher retention priority.

- We propose an information-retention token pruning architecture, named IR-Arc, where informative tokens undergo complete forward computation but uninformative tokens implement a block-level early stopping strategy, reducing storage and computation overhead effectively. IR-Arc makes TP-Spikformer exhibit high versatility and requires no training from scratch, attaining competitive performance even under zero fine-tuning conditions.

- We select a variety of architectures for our experiments, including the feature-map invariant Spikformer and SDT-V1, the feature pyramid-based QKFormer, and the advanced SDT-V3. Additionally, we assess TP-Spikformer on multiple tasks such as classification, segmentation, detection, and tracking. Through validation across multiple architectures and tasks, we demonstrate the effectiveness, efficiency, and scalability of our method.

## 2 RELATED WORK

**Spiking transformer.** Spikformer pioneers self-attention and direct Transformer training in SNNs Zhou et al., which eliminates float multiplication in attention via spike-based Q, K, and V. Spikformer V2 explores masked image modeling in spiking transformers, achieving 81.1% accuracy on ImageNet with just 1 time step Zhou et al. (2024b). SpikingResformer proposes a dual-spike self-attention and combines it with a ResNet-based architecture, improving performance with reduced parameters Shi et al. (2024). QKFormer uses spike-based Q and K for attention computation and introduces spiking patch embeddings with deformable shortcuts, achieving milestone results on multiple datasets Zhou et al. (2024a). In the SNN community, the series of Spike-driven Transformer has gained notable attention. SDT-V1 pioneers spike-driven computation in spiking transformers, converting spike-related matrix multiplications to efficient addition operations Yao et al. (2023a). SDT-V2 extends SDT-V1 into a meta architecture, exploring structure design, spike-driven attention, and skip connection to enhance performance Yao et al.. SDT-V3 optimizes spiking neuron firing patterns and designs an efficient Transformer Yao et al. (2025). Despite substantial accuracy improvements, SNNs' inherent energy efficiency is undermined, limiting their deployment in resource-limited scenarios.

**Token pruning in spiking transformer.** SparseSpikformer proposes a hybrid pruning framework operating at both weight and token levels, removing unimportant background tokens based on neurons' spike firing rates Liu et al. (2024). However, it has two limitations: it relies on firing rate for token importance without leveraging SNNs' temporal characteristics, and its validation is limited to a single architecture and small-scale datasets, leaving its scalability to other architectures and benchmarks unexplored. AT-SNN adopts an adaptive computation time (ACT) mechanism to mask unimportant tokens using Halting Scores during training, followed by a similarity-based token merging strategy to reduce computational overhead Kang et al. (2024). However, ACT introduces additional parameters requiring retraining, and AT-SNN's validation is also limited to a single architecture and simple datasets. Recently, STATA introduces an anchor token for token pruning with dual temporal and inter-layer alignment mechanisms, becoming the first token pruning method in spiking transformer validated on ImageNet Zhuge et al. (2024). However, it requires a complete retraining process, and the additional loss terms increase training overhead compared to uncompressed counterparts.

## 3 PRELIMINARY

**Spiking neuron model.** Spiking neurons mimic the information transmission and processing of biological neurons. Due to the high computational complexity of biological neurons, researchers simplify spiking neurons into differential equations for computer simulation. The neural behavior of spiking neurons typically includes three mechanisms: membrane potential integration, spike generation, and reset Wu et al. (2018); Neftci et al. (2019); Zhang et al. (2021b). Below, we describe these behaviors using the widely adopted Leaky Integrate-and-Fire (LIF) model, which can be described as follows:

$$\tilde{\mathbf{u}}^\ell[t] = \mathbf{u}^\ell[t-1] + f(\mathbf{w}^\ell, \mathbf{s}^{\ell-1}[t]), \qquad \text{(Voltage integration)}, \qquad (1)$$

$$\mathbf{s}^\ell[t] = \text{Heaviside}(\tilde{\mathbf{u}}^\ell[t] - \theta), \qquad \text{(Spike generation)}, \qquad (2)$$

$$\mathbf{u}^\ell[t] = \begin{cases} \tilde{\mathbf{u}}^\ell[t] \left(1 - \mathbf{s}^\ell[t]\right), & \text{hard reset}, \\ \tau\tilde{\mathbf{u}}^\ell[t] - \theta\mathbf{s}^\ell[t], & \text{soft reset}, \end{cases} \qquad \text{(Reset mechanism)}, \qquad (3)$$

where $\tau$ is the constant leaky factor, $t$ is the time step, $\mathbf{w}^\ell$ is the weight matrix of layer $\ell$, and $f(\cdot)$ is the convolution or linear operation followed by batch normalization (BN). As described above, neurons integrate inputs and emit a spike $\mathbf{s} \in \{0, 1\}$ when the membrane potential $\tilde{\mathbf{u}}$ exceeds the threshold $\theta$. After spike emission, the reset mechanism is invoked to update the membrane potential.

**Spiking transformer.** The spiking transformer architecture typically comprises four components: input embedding, spiking self-attention (SSA), multi-layer perceptron (MLP), and a classification head (CH). Given a 2D image sequence $I$, the input embedding module linearly projects it into $D$-dimensional spiking features vector and partitions it into either a 2D grid of $H \times W$ spiking patches or $N$ flattened spiking patches. After adding positional encoding, the initial feature $\mathbf{X}^0$ passes through $L$ transformer blocks, each containing SSA and MLP modules. Finally, the features $\mathbf{X}^L$ are

aggregated via global average pooling (GAP) and processed by the CH to generate predictions:

$$\mathbf{X}^0 = \text{InputEmbedding}\,(I)\,, \qquad\qquad \mathbf{X}^0 \in \mathbb{R}^{T \times H \times W \times D}, \qquad\qquad (4)$$

$$\hat{\mathbf{X}}^\ell = \text{SSA}(\mathbf{X}^{\ell-1}) + \mathbf{X}^{\ell-1}, \qquad\qquad \mathbf{X}^\ell \in \mathbb{R}^{T \times H \times W \times D}, \ell = 1...L, \qquad (5)$$

$$\mathbf{X}^\ell = \text{MLP}(\hat{\mathbf{X}}^\ell) + \hat{\mathbf{X}}^\ell, \qquad\qquad \mathbf{X}^\ell \in \mathbb{R}^{T \times H \times W \times D}, \ell = 1...L, \qquad (6)$$

$$\mathbf{Y} = \text{CH}(\text{GAP}(\mathbf{X}^L)). \qquad\qquad (7)$$

Notably, SSA provides an efficient approach to model the local-global information of images using spike-based queries ($\mathbf{q}$), keys ($\mathbf{k}$), and values ($\mathbf{v}$) without employing softmax, described as,

$$\text{SSA}(\mathbf{X}^{\ell-1}) = \mathcal{SN}((\mathbf{q_s}\mathbf{k_s}^\top)\mathbf{v_s}), \qquad\qquad (8)$$

$$\mathbf{x_s} = \mathcal{SN}(\mathbf{x}), \quad \mathbf{x} = \mathbf{w_x} \cdot \mathcal{SN}(\mathbf{X}^{\ell-1}), \quad \mathbf{x} \in \{\mathbf{q}, \mathbf{k}, \mathbf{v}\}. \qquad (9)$$

This design combines the efficiency of SNNs with the modeling capabilities of Transformers, enabling effective processing of information with reduced computational overhead Yao et al. (2023a).

## 4 METHOD

In this section, we introduce our simple yet effective token pruning method for efficient spiking transformers. We first present the heuristic spatiotemporal information-retaining criterion that assesses token importance. Then, we introduce the information-retention pruning architecture with the block-level early stopping strategy. By integrating these two components, our TP-Spikformer achieves improved efficiency, scalability, and effectiveness.

### 4.1 HEURISTIC SPATIOTEMPORAL INFORMATION-RETAINING CRITERION

Extensive neuroscience research has shown that the human visual system does not process all information equally, but instead prioritizes regions that are spatially salient or exhibit significant temporal changes Itti et al. (2002); Fecteau & Munoz (2006). This selective mechanism allows biological systems to allocate computational resources efficiently to informative regions of visual input Koch & Ullman (1987). Inspired by this, we propose the IRToP to guide token pruning in spiking transformers, which assesses tokens based on their information content, giving tokens with richer spatial and temporal information higher preservation priority.

**Spatial token scorer.** In the human visual system, spatial locations compete for saliency within feature maps, allowing only those that quite differ from their local surroundings to persist for further processing Itti et al. (2002). This motivates us to assess the representational divergence between each token and its spatial neighbors, assigning higher retention scores to those with greater spatial saliency.

In the spiking transformer framework, feature representations are structured either as spatial feature maps $\mathbf{X}^{\ell-1} \in \mathbb{R}^{T \times H \times W \times D}$ or in their flattened form $\mathbf{X}^{\ell-1} \in \mathbb{R}^{T \times N \times D}$, where $N = H \times W$ is the total number of tokens. Consider a spatial feature map at time step $t$, for each token located at spatial position $(h, w)$, i.e., $\mathbf{X}^{\ell-1}_{t,h,w} \in \mathbb{R}^D$, we compute the dissimilarity between this token and a representative one in its spatial window. The spatial dissimilarity of a single token is computed as,

$$\mathcal{S}_{\text{score}}(\mathbf{X}^{\ell-1}_{t,h,w}) = 1 - \frac{\langle \mathbf{X}^{\ell-1}_{t,h,w}, \mathbf{Y}^{\ell-1}_{t,h,w} \rangle}{|\mathbf{X}^{\ell-1}_{t,h,w}| \cdot |\mathbf{Y}^{\ell-1}_{t,h,w}|}, \quad \mathbf{Y}^{\ell-1}_{t,h,w} = \frac{1}{|\mathcal{W}_{h,w}|} \sum_{(p,q) \in \mathcal{W}_{h,w}} \mathbf{X}^{\ell-1}_{t,p,q}, \qquad (10)$$

where the representative token $\mathbf{Y}^{\ell-1}_{t,h,w}$ is the mean representation within the spatial window, capturing the local contextual information around the target token. Unlike pairwise similarity calculations with neighbor tokens, using this representative token reduces computational complexity. $\mathcal{W}_{h,w}$ is the set of valid neighboring positions within the $k \times k$ window centered at the coordinate $(h, w)$, defined as,

$$\mathcal{W}_{h,w} = \{(h \pm \lfloor \frac{k-1}{2} \rfloor, w \pm \lfloor \frac{k-1}{2} \rfloor)\} \cap [0, H-1] \times [0, W-1]. \qquad (11)$$

We calculate the dissimilarity for each token in the feature map at time step $t$ and normalize these values to obtain the spatial saliency score. Each score is constrained between 0 and 1, with the sum of

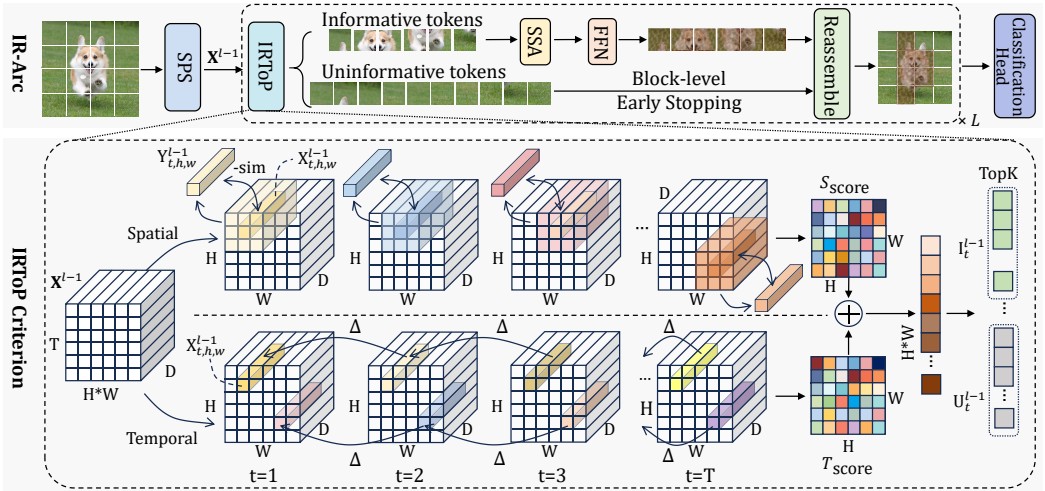

Figure 2: The overall workflow of the proposed TP-Spikformer, including the information-retention token pruning framework (**top**) and the spatiotemporal information-retaining criterion (**bottom**).

all scores equal to 1. A higher score indicates greater spatial saliency of a token, and these tokens are given higher priority for retention in token pruning. This ensures that the most spatially informative tokens of the original feature maps are preserved as much as possible during token pruning. Notably, this approach can be easily extended to frameworks with flattened token representations.

**Temporal token scorer.** Neuroscientific research has demonstrated that the human visual system is highly sensitive to sudden and significant temporal changes Rensink (2002); Nothdurft (2000). This processing mechanism serves as an efficient information compression strategy, enabling the brain to swiftly locate and process key temporal dynamics within extensive visual input. Inspired by this, we measure the temporal dynamics of each token between consecutive time steps and assign higher retention scores to tokens exhibiting significant temporal variations. Considering a token at a position $(h, w)$ in the spatial feature map, we compute its temporal variation as,

$$\mathcal{T}_{\text{score}}(\mathbf{X}_{t,h,w}^{\ell-1}) = \begin{cases} |\mathbf{X}_{t,h,w}^{\ell-1} - \mathbf{X}_{t-1,h,w}^{\ell-1}|, & \text{if } t > 1, \\ |\mathbf{X}_{t,h,w}^{\ell-1}|, & \text{if } t = 1. \end{cases} \tag{12}$$

We calculate and normalize the temporal variation of all tokens at time $t$ to obtain their temporal variation scores. Higher scores indicate richer temporal information of a token, leading to higher retention priority during pruning. This biologically inspired method allows spiking transformers to capture critical temporal features of input sequences while filtering out redundant information.

**IRToP criterion.** We combine the normalized spatial saliency and temporal variation scores for each token to obtain a spatiotemporal score at each time step, as detailed in Figure 2. Formally, for a token located at position $(h, w)$ at time step $t$, our IRToP criterion evaluates this token as follows,

$$\text{IRToP}(\mathbf{X}_{t,h,w}^{\ell-1}) = \hat{\mathcal{S}}_{\text{score}}(\mathbf{X}_{t,h,w}^{\ell-1}) + \hat{\mathcal{T}}_{\text{score}}(\mathbf{X}_{t,h,w}^{\ell-1}), \tag{13}$$

where $\hat{\mathcal{S}}$ and $\hat{\mathcal{T}}$ denotes the normalized score. Given the $\ell$-th block's pruning rate $r_\ell$, we classify tokens in the input feature map $\mathbf{X}^{\ell-1}$ into informative and non-informative tokens based on their scores. Specifically, we denote a set of token scores in $\mathbf{X}^{\ell-1}$ at time step $t$ as $\text{TS}_t^{\ell-1} = \{\text{IRToP}(\mathbf{X}_{t,h,w}^{\ell-1})\}_{h=1,w=1}^{H,W}$, so the sets of informative tokens $\mathbf{I}_t^{\ell-1}$ and uninformative tokens $\mathbf{U}_t^{\ell-1}$ is:

$$\mathbf{I}_t^{\ell-1} = \{\mathbf{X}_{t,h,w}^{\ell-1} \mid (h, w) \in \text{TopK}(\text{TS}_t^{\ell-1})\}, \tag{14}$$

$$\mathbf{U}_t^{\ell-1} = \{\mathbf{X}_{t,h,w}^{\ell-1} \mid \{(h, w)\}_{h=1,w=1}^{H,W} \notin \text{TopK}(\text{TS}_t^{\ell-1})\}, \tag{15}$$

where $\text{K} = \lceil (1 - r_\ell) \times H \times W \rceil$ is the number of tokens to retain and $\text{TopK}(\cdot)$ returns the highest-scoring token coordinates. The tokens in $\mathbf{U}_t^{\ell-1}$ are candidates for token pruning. In summary, the IRToP criterion offers a neuroscience-inspired heuristic approach for token pruning in spiking transformers, enabling computational efficiency while retaining critical information.

## 4.2 INFORMATION-RETENTION TOKEN PRUNING ARCHITECTURE

After categorizing informative and uninformative tokens based on the IRToP criterion, we propose an IR-Arc for token pruning in spiking transformers. The forward propagation of the $\ell$-th block at time step $t$ in our approach is described by the following formulas:

$$\mathbf{I}_t^{\ell-1}, \mathbf{U}_t^{\ell-1} \leftarrow \{\text{IRToP}(\mathbf{X}_{t,h,w}^{\ell-1})\}_{h=1,w=1}^{H,W}, \qquad \mathbf{X}_t^{\ell-1} \in \mathbb{R}^{H \times W \times D}, \qquad (16)$$

$$\mathbf{I'}_t^{\ell} = \text{SSA}(\mathbf{I}_t^{\ell-1}) + \mathbf{I}_t^{\ell-1}, \qquad \mathbf{I}_t^{\ell-1} \in \mathbb{R}^{k \times D}, \qquad (17)$$

$$\mathbf{X}_{t,\text{inf}}^{\ell} = \text{MLP}(\mathbf{I'}_t^{\ell}) + \mathbf{I'}_t^{\ell}, \qquad \mathbf{I'}_t^{\ell-1} \in \mathbb{R}^{k \times D}, \qquad (18)$$

$$\mathbf{X}_{t,\text{uni}}^{\ell} = \mathbf{U}_t^{\ell-1}, \qquad \mathbf{U}_t^{\ell-1} \in \mathbb{R}^{(H \times W - k) \times D}, \qquad (19)$$

$$\mathbf{X}_t^{\ell} = \text{Reassemble}(\mathbf{X}_{t,\text{inf}}^{\ell}, \mathbf{X}_{t,\text{uni}}^{\ell}), \qquad \mathbf{X}_t^{\ell} \in \mathbb{R}^{H \times W \times D}. \qquad (20)$$

In IR-Arc, informative tokens undergo complete SSA and MLP to further extract the essential features, while uninformative ones are skipped via block-level early stopping. All tokens are then reassembled into their original positions to restore the feature map size. Unlike direct token removal, IR-Arc skips the calculation of uninformative tokens and then keeps them unchanged. This not only reduces memory and computational overhead, but also retains more information during token pruning. Additionally, the retention and reassembly strategy allows TP-Spikformer to be easily extended to hierarchical spiking transformers, like QKFormer Zhou et al. (2024a), detailed in Appendix F.

We summarize the workflow of TP-Spikformer in Algorithm 1 and its advantages from two aspects. First, the information-retention strategy optimally allocates computational resources to tokens with high information content, enhancing efficiency without compromising model's feature extraction capabilities. Second, it reduces memory and computation cost by skipping uninformative tokens rather than removing them directly, ensuring compatibility with models with feature pyramids.

---

**Algorithm 1** The overall workflow of TP-Spikformer.

---

**Require:** Trained spiking transformer model; Token pruning ratio per block $r = \{r_1, \cdots, r_L\}$; Input image $I$.
**Ensure:** Classification results, with token pruning performed in forward propagation.
  ▷ $\mathbf{X}^0 \leftarrow \text{InputEmbedding}(I)$
 **for** $t \leftarrow 1$ **to** $T$ **do**
  **for** $\ell \leftarrow 1$ **to** $L$ **do**
    ▷ Get the input feature map: $\mathbf{X}_t^{\ell-1}$ and define an avg_kernel with shape $[D, D, k, k]$ and value 1;
    ▷ Get representative tokens from each $k \times k$ spatial window: $\mathbf{Y}_t^{\ell-1} \leftarrow \text{Conv2d}(\mathbf{X}_t^{\ell-1}, \text{avg\_kernel})$;
    ▷ Token scoring: $\text{TS}_t^{\ell-1} = \text{SpatialScorer}(\mathbf{X}_t^{\ell-1}, \mathbf{Y}_t^{\ell-1}) + \text{TemporalScorer}(\mathbf{X}_t^{\ell-1}, \mathbf{X}_{t-1}^{\ell-1})$;
    ▷ Select the K most informative tokens: $\mathbf{I}_t^{\ell-1} \leftarrow \{\mathbf{X}_{t,h,w}^{\ell-1} \mid (h, w) \in \text{TopK}(\text{TS}_t^{\ell-1})\}$;
    ▷ Get the remaining uninformative tokens: $\mathbf{U}_t^{\ell-1} \leftarrow \text{All Tokens} \setminus \text{Informative Tokens}$
    ▷ Extract and retain important information: $\mathbf{I'}_t^{\ell} \leftarrow \text{SSA}(\mathbf{I}_t^{\ell-1}) + \mathbf{I}_t^{\ell-1}$, $\mathbf{X}_{t,\text{inf}}^{\ell} \leftarrow \text{MLP}(\mathbf{I'}_t^{\ell}) + \mathbf{I'}_t^{\ell}$;
    ▷ Early stopping for uninformative tokens: $\mathbf{X}_{t,\text{uni}}^{\ell} \leftarrow \mathbf{U}_t^{\ell-1}$;
    ▷ Reassemble tokens to restore the original feature map size: $\mathbf{X}_t^{\ell} \leftarrow \text{Reassemble}(\mathbf{X}_{t,\text{inf}}^{\ell}, \mathbf{X}_{t,\text{uni}}^{\ell})$;
  **end for**
  ▷ Y $= \text{CH}(\text{GAP}(\mathbf{X}_t^L))$;
 **end for**

---

## 5 EXPERIMENT

In this section, we conduct extensive experiments to assess our method. **First**, we evaluate TP-Spikformer's efficacy and efficiency on various architectures and tasks, comparing it with related work and uncompressed counterparts. **Second**, we study the zero-finetuning accuracy preservation property of TP-Spikformer. **Third**, we quantify the actual efficiency gains achieved by TP-Spikformer in training time and memory cost. **Finally**, we conduct ablation studies to validate the efficacy of IRToP and IR-Arc. We also visualize temporal and spatial scores of TP-Spikformer to provide insights into their operational mechanisms. Appendix A provides details about experimental setups.

### 5.1 PERFORMANCE COMPARISON

**Image classification.** We first compare TP-Spikformer with existing SNN token pruning methods. Existing methods are mostly validated on the Spikformer and small datasets, and we compare TP-Spikformer with them in Table 1 and 2. Clearly, TP-Spikformer maintains high performance even

Table 1: Comparison of TP-Spikformer on small-scale datasets. 'S' means the method doesn't add extra parameters or need retraining. $N_{avg}$ is the average token retention ratio. $\dagger$ is an estimated token retention ratio based on reported metrics. Top three results are highlighted as first, second, and third.

| Method | S | \multicolumn{3}{c}{CIFAR-10} | | | \multicolumn{3}{c}{CIFAR-100} | | | \multicolumn{3}{c}{DVS-CIFAR10} | | |
|---|---|---|---|---|---|---|---|---|---|---|
| | | $T$ | $N_{avg}$ | Acc. (%) | $T$ | $N_{avg}$ | Acc. (%) | $T$ | $N_{avg}$ | Acc. (%) |
| Spikformer Zhou et al. | - | 4 | ×1 | 95.19 Base | 4 | ×1 | 78.21 Base | 16 | ×1 | 80.9 Base |
| SparseSpikformer Liu et al. (2024) | ✗ | 4 | ×0.85 | 95.18 (-0.01) | 4 | ×0.85 | 77.70 (-0.51) | 16 | ×0.85 | 79.3 (-1.6) |
| | ✗ | 4 | ×0.70 | 95.03 (-0.16) | 4 | ×0.70 | 77.07 (-1.14) | 16 | ×0.70 | 78.4 (-2.5) |
| | ✗ | 4 | ×0.63 | 94.77 (-0.42) | 4 | ×0.63 | 76.78 (-1.43) | 16 | ×0.63 | 79.1 (-1.8) |
| AT-SNN Kang et al. (2024) | ✗ | 4 | ×0.28 | 95.06 (-0.13) | 4 | ×0.75 | 78.14 (-0.07) | - | - | - |
| | ✗ | 4 | ×0.21 | 94.88 (-0.31) | 4 | ×0.58 | 77.27 (-0.94) | - | - | - |
| STATA Zhuge et al. (2024) | ✗ | 4 | ×0.50$^\dagger$ | 95.00 (-0.19) | 4 | ×0.50$^\dagger$ | 77.70 (-0.51) | 16 | ×0.50$^\dagger$ | 80.7 (-0.2) |
| **TP-Spikformer** | ✓ | 4 | ×0.25 | 95.16 (-0.03) | 4 | ×0.60 | 78.48 (+0.27) | 16 | ×0.78 | 81.0 (+0.1) |
| | ✓ | 4 | ×0.20 | 95.12 (-0.07) | 4 | ×0.55 | 77.83 (-0.38) | 16 | ×0.50 | 80.7 (-0.2) |

Table 2: Comparison of TP-Spikformer on ImageNet. 'Thr' reports model throughput on one A800.

| Method | Architecture | S | $T$ | $N_{avg}$ | OPs$_{block}$ (G) | Power (mJ) | Acc. (%) | Thr (imgs/s) |
|---|---|---|---|---|---|---|---|---|
| SEW Fang et al. (2021) | SEW-ResNet-152 | - | 4 | ×1 | - | 12.891 | 69.26 | - |
| Spikformer Zhou et al. | Spikformer-8-768 | - | 4 | ×1 | 18.91 | 21.48 | 74.81 | 229 |
| SNN-ViT Wang et al. (2025) | SNN-ViT-8-512 | - | 4 | ×1 | - | 35.75 | 80.23 | 378 |
| STATA Zhuge et al. (2024) | Spikformer-8-768 | ✗ | 4 | ×0.50$^\dagger$ | - | 11.16 | 74.03 | - |
| **TP-Spikformer** | SDT-V1-8-768 [*NeurIPS23*]Yao et al. (2023a) | - | 4 | ×1 | 9.04 Base | 10.26 Base | 76.32 Base | 156 Base |
| | | ✓ | 4 | ×0.74 | 6.75 (↓25%) | 8.20 (↓19%) | 75.82 (-0.50) | 181 (↑16%) |
| | | ✓ | 4 | ×0.65 | 5.93 (↓34%) | 7.46 (↓26%) | 75.62 (-0.70) | 189 (↑21%) |
| | | ✓ | 4 | ×0.51 | 4.71 (↓48%) | 6.36 (↓38%) | 74.79 (-1.53) | 202 (↑29%) |
| | QK-10-768 [*NeurIPS24*]Zhang et al. (2024) | - | 4 | ×1 | 15.08 Base | 32.12 Base | 85.56 Base | 75 Base |
| | | ✓ | 4 | ×0.72 | 10.7 (↓29%) | 28.18 (↓12%) | 84.45 (-1.11) | 84 (↑12%) |
| | | ✓ | 4 | ×0.65 | 9.61 (↓36%) | 27.19 (↓15%) | 84.32 (-1.24) | 88 (↑17%) |
| | | ✓ | 4 | ×0.53 | 7.97 (↓47%) | 25.71 (↓20%) | 82.53 (-3.03) | 106 (↑41%) |
| | SDT-V3-19M [*TPAMI25*]Yao et al. (2025) | - | 1×4 | ×1 | 1.74 Base | 5.47 Base | 79.72 Base | 1562 Base |
| | | ✓ | 1×4 | ×0.78 | 1.37 (↓21%) | 4.68 (↓14%) | 79.01 (-0.71) | 1785 (↑14%) |
| | | ✓ | 1×4 | ×0.65 | 1.13 (↓35%) | 4.43 (↓19%) | 78.10 (-1.62) | 1851 (↑19%) |
| | | ✓ | 1×4 | ×0.56 | 0.98 (↓44%) | 4.25 (↓22%) | 77.55 (-2.17) | 1886 (↑21%) |

under high compression ratios, e.g., retaining only 20% tokens on CIFAR-10 with merely a 0.07% accuracy drop. Then, we assess TP-Spikformer on Imagenet-1K and various architectures, focusing on accuracy, block operations (OPs$_{block}$), power, and throughput. As shown in Table 2, reducing tokens greatly lowers OPs$_{block}$ and power, with minimal accuracy loss. For instance, retaining only 53% tokens in QKFormer cuts OPs$_{block}$ by 47%, power by 20%, while maintaining 82.53% accuracy. This indicates TP-Spikformer serves as an effective and general token pruning method for SNNs.

**Semantic segmentation.** We use ADE20K to assess the efficacy of TP-Spikformer in semantic segmentation tasks. *This comparison aims to show TP-Spikformer's competitiveness despite token pruning, not its superiority in performance.* We use TP-Spikformer with SDT-V3 and $N_{avg}$ of 0.78 and 0.56 as the backbone for feature extraction, and other settings follow Yao et al. (2025). Results and visualizations are shown in Table 3 and Figure 3a. With only 56% tokens retained, TP-Spikformer achieves a 1.7× throughput with only a 0.2% mIoU drop compared to the uncompressed SDT-V3.

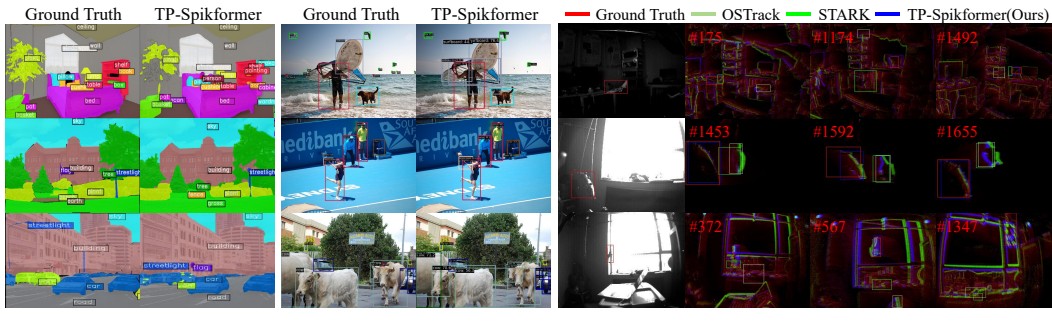

(a) Semantic segmentation     (b) Object detection     (c) Event-based object tracking

Figure 3: Visualization of ground truth and ours, showing its efficacy on diverse downstream tasks.

Table 3: Segmentation result on ADE20K.

| Method | $N_{avg}$ | $T$ | Param (M) | Thr (imgs/s) | MIoU (%) |
|---|---|---|---|---|---|
| (Yao et al.) | ×1 | 4 | 16.5 | 59.6 | 33.6 |
| | ×1 | 1 | 58.9 | 37.6 | 34.8 |
| | ×1 | 4 | 58.9 | 36.8 | 35.3 |
| (Yao et al., 2025) | ×1 | 2 | 11.0 | 82.7 | 31.9 |
| | ×1 | 4 | 11.0 | 74.5 | 40.1 |
| | ×1 | 8 | 11.0 | 70.4 | 41.4 |
| **TP-Spikformer** | ×0.78 | 4 | 11.0 | 112.2 | 40.2 |
| **TP-Spikformer** | ×0.56 | 4 | 11.0 | 128.6 | 40.0 |

Table 4: Object detection results on COCO 2017.

| Method | $N_{avg}$ | $T$ | Param (M) | Thr (imgs/s) | mAP @0.5(%) |
|---|---|---|---|---|---|
| (Kim et al., 2020) | - | 3500 | 10.2 | - | 25.7 |
| (Li et al., 2022) | - | 512 | 17.1 | - | 45.3 |
| (Zhang et al., 2023) | - | 4 | 11.3 | - | 28.5 |
| (Su et al., 2023) | - | 4 | 26.9 | - | 50.1 |
| (Yao et al.) | ×1 | 1 | 34.9 | 32.6 | 44.0 |
| (Yao et al., 2025) | ×1 | 8 | 38.7 | 29.8 | 58.8 |
| **TP-Spikformer** | ×0.78 | 4 | 38.7 | 42.9 | 55.4 |
| **TP-Spikformer** | ×0.56 | 4 | 38.7 | 43.6 | 54.4 |

**Object detection.** We use COCO2017 to evaluate the efficacy of TP-Spikformer, which also aims to show its competitiveness under token pruning instead of superior mAP. We also use SDT-V3 with $N_{avg}$ of 0.78 and 0.56 as the backbone, and others follow Yao et al. (2025). Results and visualizations are shown in Table 4 and Figure 3b. With only 78% tokens and fewer time steps, TP-Spikformer reaches a 1.4× throughput with only a 1% mAP drop, showing its efficacy in object detection.

**Event-based tracking.** We select the event-based tracking task to verify the effect of TP-Spikformer in sequence vision tasks. Experiments are conducted on three benchmarks, i.e., FE108 Zhang et al. (2021a), FELT Wang et al. (2024a), and VisEvent Wang et al. (2023). Similar in segmentation and detection, we use SDT-V3 with $N_{avg}$ of 0.78 and 0.56 as the backbone, and other settings follow Shan et al. (2025). Results and visualizations are shown in Table 3 and Figure 3c. Using only 56% of the tokens, TP-Spikformer surpasses most RGB-based trackers and rivals the advanced SDTrack, demonstrating its effectiveness in sequential vision tasks.

Table 5: TP-Spikformer vs. advanced trackers on three event-based object tracking benchmarks.

| Methods | Time step | $N_{avg}$ | Power (mJ) | FE108 AUC(%) | FE108 PR(%) | FELT AUC(%) | FELT PR(%) | VisEvent AUC(%) | VisEvent PR(%) |
|---|---|---|---|---|---|---|---|---|---|
| STARK Yan et al. (2021) | 1 | ×1 | 58.88 | 57.4 | 89.2 | 39.6 | 51.7 | 34.1 | 46.8 |
| SimTrack Chen et al. (2022) | 1 | ×1 | 93.84 | 56.7 | 88.3 | 36.8 | 47.0 | 34.6 | 47.6 |
| OSTrack$_{256}$ Ye et al. (2022) | 1 | ×1 | 98.90 | 54.6 | 87.1 | 35.9 | 45.5 | 32.7 | 46.4 |
| ARTrack$_{256}$ Wei et al. (2023) | 1 | ×1 | 174.8 | 56.6 | 88.5 | 39.5 | 49.4 | 33.0 | 43.8 |
| SeqTrack-B$_{256}$ Chen et al. (2023) | 1 | ×1 | 302.7 | 53.5 | 85.5 | 33.0 | 42.0 | 28.6 | 43.3 |
| HiT-B Kang et al. (2023) | 1 | ×1 | 19.78 | 55.9 | 88.5 | 38.5 | 48.9 | 34.6 | 47.6 |
| HIPTrack Cai et al. (2024) | 1 | ×1 | 307.7 | 50.8 | 81.0 | 38.2 | 48.9 | 32.1 | 45.2 |
| ODTrack Zheng et al. (2024) | 1 | ×1 | 335.8 | 43.2 | 69.7 | 29.7 | 35.9 | 24.7 | 34.7 |
| STNet Zhang et al. (2022) | 3 | ×1 | - | - | - | - | - | 35.0 | 50.3 |
| SDTrack$_{Tiny}$ Shan et al. (2025) | 4 | ×1 | 8.16 | 59.0 | 91.3 | 39.3 | 51.2 | 35.6 | 49.2 |
| **TP-Spikformer** | 4 | ×0.65 | 6.65 | 59.0 | 91.2 | 39.1 | 50.4 | 35.3 | 49.7 |
| **TP-Spikformer** | 4 | ×0.56 | 6.51 | 58.4 | 90.6 | 38.9 | 50.0 | 35.2 | 49.4 |

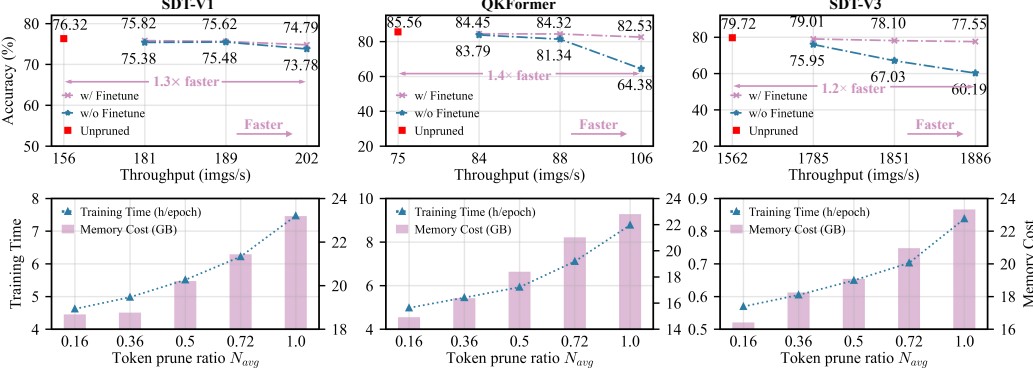

Figure 4: Zero-finetuning accuracy preservation (top) and efficiency gains (bottom) of TP-Spikformer.

Table 6: Ablation study. `Random` denotes random token pruning; `Drop` means token removal that reduces feature map size; `Spatial` and `Temporal` is using one single scorer for token selection.

| Architecture | [Random, Drop] | [Random, IR-Arc] | [Spatial, IR-Arc] | [Temporal, IR-Arc] | [IRToP, IR-Arc] |
|---|---|---|---|---|---|
| SDT-V1$_{\times 0.52}$ | 59.88% | 60.02% | 73.52% | 70.95% | 73.78% |
| QKFormer$_{\times 0.65}$ | Fail | 74.45% | 58.93% | 79.69% | 81.16% |
| SDT-V3$_{\times 0.78}$ | Fail | 73.15% | 75.95% | - | 75.95% |

## 5.2 VALIDATION AND ABLATION STUDY

**Zero-finetuning accuracy preservation of TP-Spikformer.** We observe that TP-Spikformer can also perform well in a training-free manner. To verify this, we prune three architectures using official pre-trained weights, comparing results with and without fine-tuning. As shown in Figure 4(a), TP-Spikformer attains high accuracy without fine-tuning, showing its simplicity and generalization. This makes it well-suited for real-world scenarios with limited resources and no retraining budget.

**Speedup and memory improvement of TP-Spikformer.** Besides inference throughput, we quantify the efficiency gains of TP-Spikformer in training. Experiments are conducted on ImageNet, measuring training time and memory usage. These metrics are tested on a single NVIDIA 4090, with the batch sizes of SDT-V1, QKFormer, and SDT-V3 set to 20, 15, and 200. As shown in Figure 4(b), TP-Spikformer notably reduces both training and memory cost as the token retention ratio decreases.

**Ablation study of IRToP and IR-Arc.** We conduct ablation studies on ImageNet without fine-tuning, assessing IRToP and IR-Arc. Table 6 summarizes results, with key findings outlined below. **First**, `IRToP` outperforms `Random` token pruning under `IR-Arc`, with gains of 13.76%, 6.71%, and 2.8% on SDT-V1, QKFormer, and SDT-V3, respectively, showing its efficacy. **Second**, the efficacy of `IR-Arc` is shown by comparing it with `Drop`; though their gap is small on SDT-V1 (59.88% vs. 60.02%), `IR-Arc` better supports varying feature map sizes, like QKFormer. **Third**, decoupled analyses of `Temporal` and `Spatial` show that the `Spatial` scorer suffices on SDT-V1, while `Temporal` is more important on QKFormer, showing that both scorers in IRToP are indispensable.

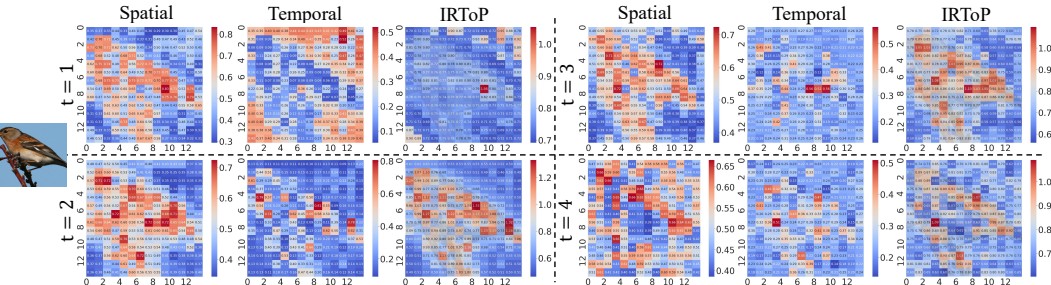

Figure 5: Visualization of spatial and temporal token scores in the 8th block of SDT-V1-8-768.

**Decoupling analysis of IRToP.** We decouple and visualize spatial and temporal scores to understand their roles. Figure 5 shows token scores from the last block before the classification head, where the spatial scorer assigns higher scores to tokens related to the main subject, while the temporal one emphasizes edges and key parts (e.g., claws, wings, beak). We find the temporal scorer underperforms at time step 1, likely due to the large magnitudes of background tokens causing misidentification. This also explains the poor token pruning results at the first time step in Figure 1. We thus recommend using only spatial scores at the first step and combining both from the second step onward.

## 6 CONCLUSION

Existing transformer-based SNNs integrate transformer performance with SNN efficiency, yet are constrained by increased model size and computational demands. This paper presents TP-Spikformer, a simple yet effective token pruning approach for spiking transformers that reduces memory and computation overhead. Drawing inspiration from human visual processing, TP-Spikformer implements the IRToP criterion and IR-Arc architecture, striking an excellent balance between efficiency and performance across multiple architectures and tasks. Extensive experiments and comprehensive studies demonstrate its value in real-world scenarios with limited resources and no retraining budget.

## ACKNOWLEDGMENTS

This work was supported by the National Natural Science Foundation of China (Grants 62576080 and 62220106008), the Sichuan Science and Technology Program (Grant 2024NSFTD0034), the Guangdong Introducing Innovative and Entrepreneurial Teams (Grant 2023ZT10×044), and the Shenzhen Science and Technology Research Fund (Grant JCYJ20220818103001002).

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

## A  EXPERIMENT DETAILS

### A.1  IMAGE CLASSIFICATION

**Dataset**   We evaluate TP-Spikformer on both static and dynamic datasets. The static datasets include CIFAR-10, CIFAR-100, and the large-scale ImageNet-1K. CIFAR-10 is a widely used benchmark in computer vision, containing 10 categories with 6,000 32×32 images per category Krizhevsky et al. (2009). CIFAR-100 maintains the same image size but expands the categories to 100, which are organized into 20 superclasses Krizhevsky et al. (2009). ImageNet-1K is a large-scale vision dataset, featuring approximately 1.28 million training images and 50,000 test images across 1,000 categories Deng et al. (2009). Its diverse categories and rich image content make it a critical benchmark for image classification. Additionally, DVS-CIFAR10 is a dynamic dataset derived from CIFAR-10 using a dynamic vision sensor Li et al. (2017). This dataset includes 9,000 training samples and 1,000 test samples, with temporal resolution in the microsecond range and spatial resolution of 128×128.

**Experimental setup**   For small-scale datasets CIFAR and DVS-CIFAR-10, we perform experiments with two token pruning ratio settings. Specifically, $N_{avg}$ is set to 0.25 and 0.20 for CIFAR-10, and 0.65 and 0.55 for CIFAR-100. We use the Spikformer-4-384 as used in studies Liu et al. (2024); Kang et al. (2024); Zhuge et al. (2024), fine-tuning with a learning rate of 5e-5 and time step 4. For DVS-CIFAR10, we set $N_{avg}$ to 0.78 and 0.5, using the Spikformer-2-384 and fine-tuning with a learning rate of 7e-4. Other experimental settings follow prior work.

Table 7: Token preserving ratio on small datasets.

| Block | CIFAR-10 | | CIFAR-100 | | DVS-CIFAR10 | |
|---|---|---|---|---|---|---|
| | **0.25** | **0.20** | **0.60** | **0.55** | **0.78** | **0.50** |
| 1 | 0.76 | 0.56 | 0.76 | 0.76 | 0.78 | 0.76 |
| 2 | 0.14 | 0.14 | 0.76 | 0.76 | 0.78 | 0.25 |
| 3 | 0.06 | 0.06 | 0.76 | 0.56 | - | - |
| 4 | 0.06 | 0.06 | 0.14 | 0.14 | - | - |

For the large-scale ImageNet, we conduct experiments with three pruning ratios for each structure, detailed in Table 2. We summarize the token retention ratio per block of small datasets in Table 7 and Imagenet-1K in Table 8. In the next paragraph, we introduce how the token retention ratio per block is obtained. During fine-tuning, we remove the warm-up epoch and set the learning rate to 1e-5 for 50 epochs. The batch size per GPU for SDT-V1 and QKFormer is 72, while 760 for SDT-V3. The detailed settings for ImageNet are shown in Table 9. In all experiments, when calculating the spatial scores of tokens, we compute the set of neighboring positions in a $3 \times 3$ window, i.e., $k = 3$. We also discuss the effect of $k$ on performance in C.

Table 8: Token preserving ratio in each transformer block on Imagenet-1K.

| Block | SDT-V1 | | | QKFormer | | | SDT-V3 | | |
|---|---|---|---|---|---|---|---|---|---|
| | **0.74** | **0.65** | **0.51** | **0.72** | **0.65** | **0.53** | **0.78** | **0.65** | **0.56** |
| 1 | 1 | 1 | 0.73 | 0.90 | 0.81 | 0.64 | 1 | 0.81 | 0.64 |
| 2 | 1 | 0.73 | 0.73 | 0.90 | 0.81 | 0.64 | 0.90 | 0.81 | 0.64 |
| 3 | 0.73 | 0.73 | 0.51 | 0.72 | 0.64 | 0.64 | 0.90 | 0.64 | 0.64 |
| 4 | 0.73 | 0.73 | 0.51 | 0.72 | 0.64 | 0.49 | 0.90 | 0.64 | 0.64 |
| 5 | 0.73 | 0.51 | 0.51 | 0.72 | 0.64 | 0.49 | 0.64 | 0.64 | 0.49 |
| 6 | 0.73 | 0.51 | 0.51 | 0.72 | 0.64 | 0.49 | 0.64 | 0.56 | 0.49 |
| 7 | 0.51 | 0.51 | 0.32 | 0.64 | 0.64 | 0.49 | 0.64 | 0.56 | 0.49 |
| 8 | 0.51 | 0.51 | 0.32 | 0.64 | 0.56 | 0.49 | 0.64 | 0.56 | 0.49 |
| 9 | - | - | - | 0.64 | 0.56 | 0.49 | - | - | - |
| 10 | - | - | - | 0.64 | 0.56 | 0.49 | - | - | - |

In order to find the optimal pruning combination between blocks, we employ a search strategy before fine-tuning to determine the token pruning ratio per block based on the given global pruning rate. The grid search used is a very simple method, which is intended to perform a coarse search to initially identify a reasonable pruning rate. Given a pre-trained model and a global token preservation, we summarize its detailed search process below.

Table 9: Experimental setups on Imagenet-1K.

| Hyper-parameter | SDT-V1 | QKFormer | SDT-V3 |
|---|---|---|---|
| $N_{\text{avg}}$ | 0.74, 0.65, 0.51 | 0.72, 0.65, 0.51 | 0.78, 0.65, 0.56 |
| $k$ in IRToP | 3 | 3 | 3 |
| Time step | 4 | 4 | 4 |
| Warmup epoch | None | None | None |
| Epoch | 50 | 50 | 50 |
| Resolution | 224×224 | 224×224 | 224×224 |
| Batch size per GPU | 72 | 72 | 760 |
| Optimizer | Adam | Adam | Adam |
| Weight decay | 0 | 0 | 0 |
| Initial learning rate | 1e-5 | 1e-5 | 1e-5 |
| Learning rate decay | Cosine | Cosine | Cosine |

- First is to obtain a set of token preservation ratio combinations. Specifically, the search space for ratios is restricted to a small set of discrete values, such as 0.9×0.9, 0.8×0.8, 0.75×0.75, 0.6×0.6, etc. Furthermore, we impose a monotonic constraint, requiring the token preservation ratio to decrease progressively from shallow to deeper blocks. This is motivated by the observation that shallow layers capture low-level features and thus require higher token retention, while deeper layers handle high-level semantic information and can tolerate more aggressive token pruning Lin et al. (2021).

- Second, we randomly sample a small batch of data from the training dataset and evaluate the accuracy for each combination in the combinations set. The combination with the top-1 accuracy is selected and used for subsequent fine-tuning.

**By reviewing the search logs, we observe that different configurations give similar performance under the same global ratio. Therefore, the grid search is only used for SDT-V3, while for QKFormer and SDT-V1, we directly set the ratios manually and fine-tune the models without performing grid search.** We summarize the search details for SDT-V3 in Table 10, including the discrete search space, the number of candidate combinations, the time to evaluate each combination, and the total search time.

Table 10: Search details for the SDT-V3.

| Model | $N_{avg}$ | Searching space per token retention ratio | Number of combinations | Evaluation time per combination | Total Time (4*NVIDIA 4090) |
|---|---|---|---|---|---|
| SDT-V3 | 0.78 | [1,0.90,0.81,0.72,0.64,0.56,0.49] | 65 | 26s | 28min 11s |
| SDT-V3 | 0.65 | [0.81,0.72,0.64,0.56,0.49,0.42,0.36] | 89 | 22s | 32min 38s |
| SDT-V3 | 0.56 | [0.81,0.72,0.64,0.56,0.49,0.42,0.36] | 166 | 16s | 44min 16s |

## A.2 SEMANTIC SEGMENTATION

**Dataset** ADE20K Zhou et al. (2019) is a widely used and well-established dataset for semantic segmentation in computer vision research. It comprises approximately 25,000 images, with over 20,000 images designated for training, 2,000 images for validation, and 3,000 images for testing. Each image in the dataset is densely annotated with pixel-level labels across 150 distinct semantic categories. These categories cover a wide array of objects, such as people, cars, and animals, as well as scene elements like sky, roads, and vegetation, each with intricate visual features that make semantic segmentation tasks more challenging. Due to its diversity and complexity, ADE20K serves as a critical and challenging benchmark for evaluating the performance of segmentation algorithms.

**Experimental setup** In this work, we begin by converting the *mmsegmentation* Contributors (2020) codebase to its spike-based version, inspired by the SDT-V3 Yao et al. (2025). We employ TP-Spikformer with SDT-V3-19M as the backbone for feature extraction, integrated with spike FPN

(Kirillov et al., 2019) for segmentation. The backbone is initialized using pretrained weights from ImageNet, ensuring that the network has a strong starting point for feature extraction. The newly added layers are initialized using the Xavier method Glorot & Bengio (2010). The experimental settings follow the parameters set in SDT-V3 to ensure consistency and comparability. We fine-tune the model with two ratios same as object detection on 4×4090 with a batch size of 12 per GPU, while original SDT-V3 is limited to 8. The results in Table 3 show that our method maintains the performance of SDT-V3 and greatly increases throughputs. This comparison is not intended to demonstrate that our method achieves top-1 accuracy, but rather to highlight that our approach remains competitive even under token pruning conditions.

### A.3 OBJECT DETECTION

**Dataset**   We evaluate TP-Spikformer on COCO2017 Lin et al. (2014), a large-scale benchmark that is widely used for object detection tasks. The dataset comprises a total of 118K training images, 5K validation images, and 40K test images, providing a comprehensive and diverse set of visual data. It covers 80 object categories, including everyday items such as cars, bicycles, animals, and household objects, which are essential for testing the algorithm's ability to recognize and interpret a wide range of visual content. In addition to object categories, COCO offers multiple types of annotations, including object instance segmentation masks, keypoints, and captions, all of which contribute to the dataset's robustness for evaluating various vision tasks. Notably, COCO emphasizes contextual relationships between objects within complex, everyday scenes, offering a more realistic and challenging evaluation setting compared to simpler datasets. This makes COCO a crucial benchmark for assessing the performance of computer vision algorithms in real-world, practical applications.

**Experimental setup**   Similar to semantic segmentation, we begin by converting the *mmdetection* Chen et al. (2019) codebase to its spike-based version. Our model architecture integrates TP-Spikformer with Mask R-CNN (He et al., 2017). The backbone is initialized using pretrained weights from ImageNet, and the newly added layers are initialized using the Xavier method (Glorot & Bengio, 2010). We fine-tune the model with two different average token retention ratios: 0.56 and 0.78. These two settings allow us to explore how different levels of token retention influence the model's performance in object detection and segmentation tasks. The experiments are conducted on a 4×A800 setup, with a batch size of 5 per GPU, providing ample computational resources to handle the large-scale training process. The results are summarized in Table 4, which also highlights that our approach remains competitive in object detection tasks.

### A.4 EVENT-BASED TRACKING

**Dataset**   We use three event-based tracking benchmarks to assess our TP-Spifkormer, detailed as:

- FE108 is captured by the DAVIS346 dynamic vision sensor, with an event rate spanning a range from 0 to 3800 events/ms Zhang et al. (2021a). This dataset features 21 diverse target categories. The diversity of categories and the high event rate make FE108 particularly useful for evaluating event-based models under varying conditions.
- FELT Wang et al. (2024a) is specifically designed to address the challenges associated with long-term object tracking in dynamic environments. This dataset places a strong emphasis on scenarios where the loss and recovery of targets are crucial for maintaining tracking accuracy.
- VisEvent Wang et al. (2023) is a large-scale dataset dedicated to event-based visual tasks, offering a robust testing ground for various event-driven models and algorithms under extreme conditions. With its broad scope, VisEvent includes a wide range of event-based visual tasks, providing a unique and challenging environment for assessing model performance.

These datasets serve as three of the most important benchmarks in event-based tracking. Their diversity makes them important for evaluating the performance of event-driven models and algorithms.

**Experimental setup**   We use the SDTrack pipeline to build a tracker for event-based tracking tasks Shan et al. (2025). Specifically, we train the tracker using an image pair matching task Chen et al.

(2022); Yan et al. (2021); Ye et al. (2022) and employ weighted focal loss Law & Deng (2018) for classification. For the predicted bounding boxes, L1 loss and generalized IoU loss Rezatofighi et al. (2019) are used for bounding box regression. We train the model for 100 epochs on the FE108 and VisEvent datasets, using a pretrained ImageNet-1K model, and for 300 epochs on the FELT dataset. For each training epoch on the FE108 and FELT datasets, we randomly sample 60,000 sample pairs with a maximum interval of 200, while on VisEvent, we use 30,000 pairs. The learning rate used during training is 4e-4, decaying to 4e-5 at 80% of the training progress. We apply normalization and regularization on the FELT dataset, and a Hanning window penalty is used to constrain the predicted boxes. However, no data augmentation or preprocessing is applied to the FE108 and VisEvent. All of the above experimental settings are strictly aligned with SDTrack Shan et al. (2025).

## B    MEASUREMENT OF EFFICIENCY METRICS

Table 11: Training time and memory usage of TP-Spikformer under different ratios and architectures.

| $N_{avg}$ | SDT-V1 | | QKFormer | | SDT-V3 | |
| --- | --- | --- | --- | --- | --- | --- |
| | Memory usage (GB) | Training time (h/epoch) | Memory usage (GB) | Training time (h/epoch) | Memory usage (GB) | Training time (min/epoch) |
| ×1 | 23.19 | 7.47 | 22.80 | 8.78 | 23.33 | 50.27 |
| ×0.72 | 21.44 | 6.22 | 21.03 | 7.11 | 20.96 | 42.16 |
| ×0.50 | 20.21 | 5.51 | 18.40 | 5.93 | 19.08 | 38.96 |
| ×0.36 | 18.76 | 4.98 | 16.37 | 5.45 | 18.25 | 36.29 |
| ×0.16 | 18.68 | 4.62 | 14.92 | 4.98 | 16.42 | 34.16 |

As shown in Figure 4, we report the training time per epoch, GPU memory usage, and inference throughput for SDT-V1, QKFormer, and SDT-V3. In this section, we provide a detailed description of how these metrics are measured, including the measurement method, experimental setup, and results.

The training time per epoch and GPU memory usage are measured by monitoring the time taken to complete one epoch and the GPU memory consumption during training. These metrics are tested on a single NVIDIA 4090, with batch sizes fixed at 20 for SDT-V1, 15 for QKFormer, and 200 for SDT-V3. We measure these two metrics of TP-Spikformer under different token pruning rates, with the results summarized in Table 11. It is evident that the proposed TP-Spikformer significantly reduces both training time and memory consumption, resulting in substantial efficiency gains. As a result, under the same configuration, when maximizing GPU resource utilization, TP-Spikformer typically allows for a higher batch size compared to its uncompressed counterpart.

For the metric of inference throughput, we estimate it by calculating the number of images processed per second during the inference process. The throughput values reported in Figure 4 and Tables 2-4 are measured on a single NVIDIA A800 GPU, with a batch size of 36 for both SDT-V1 and QKFormer, and 1024 for SDT-V3 in the classification task. In the case of segmentation and detection tasks, the batch size is fixed to 1. The obtained throughput results, presented in Tables 2 to 4, display that TP-Spikformer achieves higher throughput across all models while maintaining high accuracy. This enhances the efficiency of model inference and faster real-time processing, making it more suitable for deployment in resource-constrained scenarios that require real-time processing.

## C    ANALYSIS OF $k$ IN THE IRTOP CRITERION

In this section, we evaluate the impact of different values of $k$ on the performance of TP-Spikformer, specifically SDT-V1, QKFormer, and SDT-V3, on the ImageNet-1K dataset under zero-finetuning conditions. The results are summarized in Table 12. Each architecture shows stable performance across the different values of $k$, with a slight decrease in accuracy as $k$ increases. This drop in performance may be attributed to the fact that larger spatial windows, while capturing more global context, reduce focus on important local details, which is crucial for tasks like fine-grained detection and segmentation. Therefore, we select $k = 3$ for the experiments presented in the main text, as it offers a good balance between computational efficiency and effectiveness.

Table 12: Performance of TP-Spifkormer with different $k$ on ImageNet-1K without finetuning.

| $k$ value | SDT-V1 | | | QKFormer | | | SDT-V3 | | |
|---|---|---|---|---|---|---|---|---|---|
| | **0.74** | **0.65** | **0.51** | **0.72** | **0.65** | **0.53** | **0.78** | **0.65** | **0.56** |
| 3 | 75.38 | 75.48 | 73.78 | 83.79 | 81.34 | 64.38 | 75.95 | 67.03 | 60.19 |
| 5 | 75.37 | 75.48 | 73.77 | 83.65 | 81.23 | 64.18 | 75.77 | 66.59 | 59.71 |
| 7 | 75.34 | 75.48 | 73.77 | 83.66 | 81.20 | 64.32 | 75.59 | 66.47 | 59.40 |

## D ADAPTIVE WEIGHTING OF SPATIAL AND TEMPORAL SCORERS IN IRToP

As for Eq. 13, we have explored adaptive weighting between spatial and temporal components to analyze the effects of adaptive weight in the IRToP criterion. Specifically, we introduce a learnable parameter $\alpha$ to balance the spatial and temporal scores adaptively. The modified IRToP criterion is formulated as:

$$\text{IRToP}(\mathbf{X}_{t,h,w}^{\ell-1}) = \alpha \times \hat{\mathcal{S}}_{\text{score}}(\mathbf{X}_{t,h,w}^{\ell-1}) + (1-\alpha) \times \hat{\mathcal{T}}_{\text{score}}(\mathbf{X}_{t,h,w}^{\ell-1}), \tag{21}$$

where $\alpha$ is initialized to 0.5 (equal weighting) and is differentiable, allowing it to be optimized during training. This enables the model to automatically learn the optimal balance between spatial and temporal importance. We conduct experiments on ImageNet-1K using SDT-V1 with a pruning ratio of 0.51, fine-tuning for 50 epochs. The results are summarized Table 13. Compared to fixed equal weighting ($\alpha = 0.5$), we observe a slight performance drop when using the adaptive weighting method. We suspect this is because the learned value of $\alpha$ converges to around 0.3, suggesting that the model tends to emphasize temporal features over spatial ones. In this case, the model may overly focus on critical temporal dynamics and local details, potentially at the expense of broader spatial context that is essential for robust feature representation, leading to the observed performance degradation.

Table 13: Analysis of adaptive weighting of spatial and temporal scorers in IRToP.

| Model | Ratio | Final ratio $\alpha$ | Fine-tuning accuracy under adaptive $\alpha$ | Fine-tuning accuracy under fixed $\alpha$=0.5 |
|---|---|---|---|---|
| SDT-V1 | 0.51 | 0.3 | 74.23% | 74.79% |

## E DETAILED ANALYSIS OF IRToP

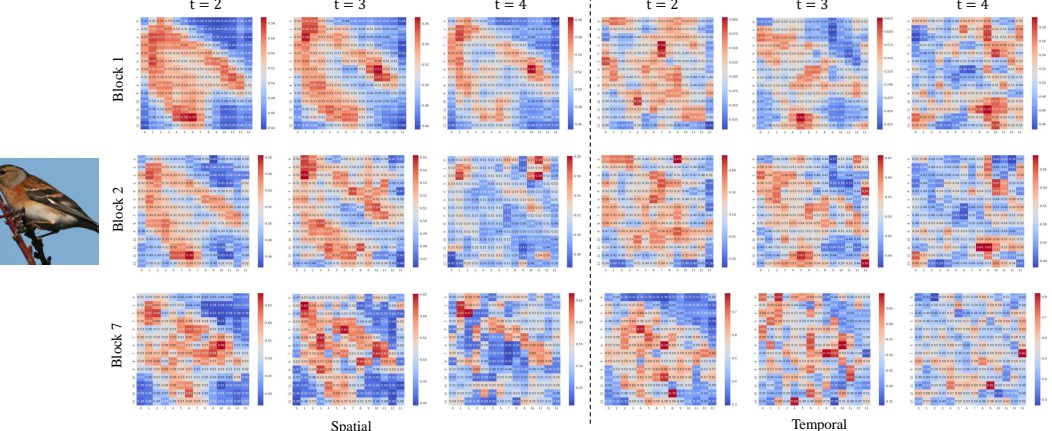

Figure 6: Detailed visualization of spatial and temporal token scores of SDT-V1-8-768.

IRToP proves effective by selecting informative tokens with two key characteristics: tokens that represent the overall outline and tokens that capture specific detail features. As a supplement to

Figure 5, we further analyze the scores of tokens selected from the 1st, 2nd, and 7th encoder blocks. In Figure 6, we decouple and visualize their spatial and temporal scores. From the spatial scores, it is clear that, irrespective of the time step and block, the spatial token scorer assigns higher scores to tokens representing the main outline, followed by those representing specific details, and lastly, background tokens. In terms of temporal scores, the temporal token scorer further extracts feature information by assigning higher scores to tokens that correspond to specific detail features. For example, in the 1st block at $t = 3$, tokens representing the bird's claws receive high scores. In the 2nd and 7th blocks at $t = 3$, tokens representing the head, wings, and branches are assigned high scores. Similarly, at $t = 4$ in the 7th block, tokens representing the bird's tail are given high scores.

The above visualization results can be explained by the design of the spatial token scorer and the temporal token scorer. The spatial token scorer relies on the similarity with neighboring tokens, assigning higher scores to tokens that differ significantly from their local surroundings. This is why the spatial token scorer extracts more information representing texture and boundary features. The computation of the temporal token scorer is inspired by the working mechanism of the human visual system. Specifically, tokens representing distinct features are captured over time, while background tokens are gradually ignored. As a result, the temporal token scorer is able to extract specific feature information. Overall, by combining the spatial token scorer with the temporal token scorer, IRToP effectively selects informative tokens, reducing computational resources while maintaining performance.

## F WHY CANNOT DIRECT TOKEN PRUNING BE EXTENDED TO THE FEATURE VARIANT SPIKING TRANSFORMER?

Direct token pruning refers to identifying uninformative tokens and discarding them without further processing in the subsequent network layers. In this section, we first discuss why this approach cannot be applied to feature-variant spiking transformers. Then, we use a feature-variant QKFormer as an example to illustrate the issue. Finally, we explain how our method effectively solves this issue.

Early SNN transformers, e.g., Spikformer Zhou et al. and SDT-V1 Yao et al. (2023a), follow ViT-style designs from ANNs, using patch embedding and standard transformer blocks. As the field developed, recent SOTA models like QKFormer Zhou et al. (2024a) and SDT-V2/V3 Yao et al.; 2025) incorporate convolution layers with kernels larger than 1 inside transformer blocks. For example, QKFormer applies conv-based Spiking Patch Embedding before each block, and SDT-V3 uses spike-based separable convolutions before every attention layer. Unlike ANNs where features can be flattened for token pruning, these convolutional layers embedded in the transformer blocks require structured and square feature maps for token pruning in SNNs. These feature-variant spiking transformers include many operations that reduce the size of feature maps, such as downsampling and convolution. The inherent structural sensitivity of these operations makes direct token pruning incompatible with feature-variant spiking Transformers, which can be understood from two aspects:

- On the one hand, convolution operations rely on structured grid-like inputs. This means that if tokens are removed from a transformer block, the remaining tokens may no longer form a valid image layout, making them incompatible with later convolutional layers.

- On the other hand, due to its strong prior assumptions, e.g., spatial local correlation and translation invariance, the convolution operation heavily relies on the spatial structure of feature maps. However, removing tokens disrupts the spatial structure of feature maps. This disruption (1) impairs local information propagation, (2) degrades the effectiveness of trained filters, and (3) compromises the model's representational ability.

As detailed in the above two reasons, it is the existence of convolution in spiking transformers that makes direct token removal infeasible, while our block-level early stopping strategy remains viable. This also indicates that, when pruning advanced spiking transformers like QKFormer and SDT-V2/V3, it is essential to preserve the overall architectural integrity. Notably, existing SNN token pruning methods have only been tested on ViT-like Spikformer and spike-driven transformer V1, and have not yet been applied to recent SOTA spiking transformers. To the best of our knowledge, we are the first to evaluate token pruning on these advanced spiking architectures.

We then use the advanced hierarchical transformer architecture QKFormer in SNNs as an example to illustrate the above issue. In QKFormer, each stage consists of the Spiking Patch Embedding with Deformed Shortcut (SPEDS) module and QKFormer block. The SPEDS module includes structure-sensitive convolution and pooling operations, which reduce the number of tokens by a $2 \times 2$ patch size before each stage and transform the number of channels into $2C$ to generate hierarchical spiking representations. If we directly prune the uninformative tokens identified by IRToP criterion in the first stage, the remaining informative tokens need be reorganized into a new feature map before being input into the second stage. This will lead to the following two challenges.

- *Difficulty in reshaping the feature map*. The reorganizing process typically requires the remaining informative tokens to be arranged into a square feature map for efficient processing in the next stage (e.g., 196 = 14×14). However, after pruning uninformative tokens in the first stage, the number of remaining tokens often cannot form the required square shape for reshaping the feature map.

- *Disruption of spatial structure*. Even if we constrain the remaining informative tokens' count to match the square shape, the spatial structure of the reconstructed feature map is inevitably disrupted. This results in the failure of well-trained parameters in the convolution operations within the SPEDS module of the second stage. This would affect the information flow in subsequent network layers and significantly degrade model performance.

TP-Spikformer addresses this challenge by introducing a block-level early stopping strategy for uninformative tokens. Instead of directly removing tokens that would disrupt the spatial structure of the feature map, TP-Spikformer bypasses the processing of uninformative tokens within the transformer blocks, and then reorganizes all tokens spatially before inputting them into the next stage. This process reduces the memory and computational overhead associated with token pruning by bypassing the computation of uninformative tokens. Moreover, by preserving the integrity of the feature map's spatial structure, TP-Spikformer avoids the difficulties of reshaping tokens and maintains the well-trained parameters of the filters, ensuring that the model maintains competitive performance even without fine-tuning.

## G  ZERO-FINETUNING ACCURACY PRESERVATION OF TP-SPIKFORMER

The zero-finetuning accuracy preservation of TP-Spikformer is made in a comparative sense. Existing advanced token pruning methods in SNNs often modify the original model architecture when applied to spiking transformers. These modifications may include introducing new tokens (STATA Zhuge et al. (2024)), adding trainable modules (ACT Kang et al. (2024)). Since these additions are randomly initialized, they require full retraining, which significantly increases data requirements, training costs, and reduces generalizability. Therefore, though our method does not completely preserve accuracy on QKFormer and SDT-V3, it achieves better accuracy than existing spiking token pruning methods under the same no-fine-tuning setting.

In Section 5.2, we conduct an in-depth analysis of TP-Spikformer's zero-finetuning performance preservation property in image classification tasks. In this section, we demonstrate that TP-Spikformer exhibits this property in other vision tasks as well. We evaluate the performance of the unpruned model using publicly available SDT-V3 detection and segmentation code and obtain results of 39.49% MIoU for semantic segmentation and 53.9% mAP@0.5 for object detection, respectively. Then, we directly apply our token pruning method to the obtained weights without any fine-tuning. With a compression ratio of 0.78, TP-Spikformer achieves performance of

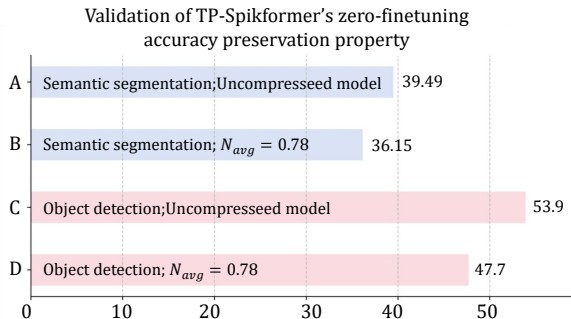

Figure 7: TP-Spikformer's zero-finetuning accuracy preservation on segmentation and detection.

36.15% and 47.7% in semantic segmentation and object detection, representing reductions of 3.34% and 6.2%, respectively. These results indicate that TP-Spikformer maintains its zero-finetuning

performance preservation property in various downstream vision tasks, highlighting its effectiveness for real-world scenarios with limited resources and no retraining budget.

## H    EFFECT OF TP-SPIKFORMER IN ACCELERATING TRAINING FROM SCRATCH

Table 14: Effect of TP-Spikformer in the training process.

| Model | Params (M) | Token preservation ratio | Total training time | GPU Memory (batch size=1280) | Acc. |
|---|---|---|---|---|---|
| SDT-V3 | 5.1 | 1 | 57h14min | 77.49 GB | 73.9% |
| TP-Spikformer | 5.1 | 0.65 | 50h 49min | 66.90 GB | 73.6% |

Although our method is mainly designed to improve deployment efficiency, it can also be easily applied during training. To verify this, we conduct experiments on large-scale ImageNet using SDT-V3-5M with and without our token pruning method, both trained from scratch. All experiments are run on a single H800 GPU, using the same settings as the original SDT-V3 to ensure a fair comparison. As shown in Table 14, TP-Spikformer achieves 73.6% accuracy, close to the 73.9% of the unpruned model, while reducing training time by 7.5 hours. This shows its effectiveness in speeding up training. Moreover, TP-Spikformer uses much less GPU memory due to fewer tokens, which allows larger models or batch sizes to be trained on the same hardware.

## I    VALIDITY OF TP-SPIKFORMER ON NON-VISUAL TASKS

The vision datasets used in the manuscript, like ImageNet, COCO, and ADE20K, are universally recognized as complex datasets in the fields of classification, detection, and segmentation, respectively. These experimental results prove the effectiveness of $S^2NN$ in complex image tasks. To further show the efficacy of our method in non-image tasks, we have conducted experiments in NLP tasks.

We extend our TP-Spikformer to the SpikeLM proposed by Xing et al. (2024) without additional architecture adjustment. Experiments utilize a 12-layer BERT-based encoder transformer and are performed on the GLUE benchmark. The token preservation ratio is set to [1, 1, 1, 0.9, 0.9, 0.9, 0.8, 0.8, 0.8, 0.7, 0.7, 0.7], while all other training configurations follow the original paper. The results are presented in the table below. Clearly, TP-Spikformer achieves a performance of 75.9%, showing no significant loss. These results confirm the effectiveness of our method on NLP tasks, demonstrating its general applicability beyond the vision domain.

Table 15: Validation of TP-Spikformer on the GLUE benchmark.

| Model | SST-2 | MRPC | RTE | MNLI | QNLI | QQP | CoLA | STS-B | Avg. |
|---|---|---|---|---|---|---|---|---|---|
| SpikeLM | 87.0 | 85.7 | 69.0 | 77.1 | 85.3 | 83.9 | 38.8 | 84.9 | 76.5 |
| TP-Spikformer | 87.9 | 84.7 | 68.2 | 76.0 | 84.6 | 84.2 | 37.0 | 84.9 | 75.9 |

## J    POTENTIAL OF TP-SPIKFORMER WITH OTHER COMPRESSION TECHNIQUES

Our TP-Spikformer is orthogonal to other lightweight approaches and can be used in conjunction with them. Here, we investigate the combination of TP-Spikformer with quantization. We select the Q-SDT proposed by Qiu et al. (2025) to evaluate this combination. Specifically, we conduct experiments on CIFAR-10 by applying TP-Spikformer to Q-SDT, training the model from scratch. The results are shown in Table 16. While there is a performance gap compared to the baseline, the 96.9% accuracy demonstrates that our token pruning method can work effectively with quantization techniques.

Table 16: Potential of TP-Spikformer with quantization.

| Model | Preserving Ratio | Bits | GPU Memory (GB) | Accuracy (%) |
|---|---|---|---|---|
| Q-SDT Qiu et al. (2025) | 1 | 4-1 | 19.27 | 97.8% |
| Q-SDT+TP-Spikformer | 0.54 | 4-1 | 15.98 | 96.9% |

# K  LEARNING ALGORITHM OF TP-SPIKFORMER

We employ the widely used spatiotemporal backpropagation (STBP) Wu et al. (2018); Jiang et al. (2025) to train the TP-Spikformer, where we need to compute gradients of the loss function $\mathcal{L}$ to synaptic weights. Through chain rule decomposition, we can decompose this gradient as,

$$\frac{\partial \mathcal{L}}{\partial \mathbf{w}_{ij}^\ell} = \sum_{t=1}^{T} \left( \frac{\partial \mathcal{L}}{\partial \mathbf{s}_i^\ell[t]} \frac{\partial \mathbf{s}_i^\ell[t]}{\partial \tilde{\mathbf{u}}_i^\ell[t]} \frac{\partial \tilde{\mathbf{u}}_i^\ell[t]}{\partial \mathbf{w}_{ij}^\ell} + \frac{\partial \mathcal{L}}{\partial \mathbf{u}_i^\ell[t+1]} \frac{\partial \mathbf{u}_i^\ell[t+1]}{\partial \tilde{\mathbf{u}}_i^\ell[t]} \frac{\partial \tilde{\mathbf{u}}_i^\ell[t]}{\partial \mathbf{w}_{ij}^\ell} \right), \tag{22}$$

where the derivative of the loss function with respect to the spike and membrane potential, i.e., $\partial \mathcal{L}/\partial \mathbf{s}_i^\ell[t]$ and $\partial \mathcal{L}/\partial \mathbf{u}_i^\ell[t+1]$ are obtained iteratively, the terms of $\partial \tilde{\mathbf{u}}_i^\ell[t]/\partial \mathbf{w}_{ij}^\ell$, $\partial \mathbf{u}_i^\ell[t+1]/\partial \tilde{\mathbf{u}}_i^\ell[t]$, and $\partial \tilde{\mathbf{u}}_i^\ell[t]/\partial \mathbf{w}_{ij}^\ell$ can be calculated based on Eq. 1. Unfortunately, a fundamental challenge in this training arises from the non-differentiable nature of spike emission. Mathematically, the gradient of the spike generation function as described in Eq. 2, i.e., $\partial \mathbf{s}_i^\ell[t]/\partial \tilde{\mathbf{u}}_i^\ell[t]$, becomes undefined at the firing threshold $\theta$ and vanishes elsewhere. This discontinuity prevents the direct application of standard backpropagation algorithms commonly used in deep learning. To overcome this limitation, we use surrogate gradient functions to approximate the derivative of $\partial \mathbf{s}_i^\ell[t]/\partial \tilde{\mathbf{u}}_i^\ell[t]$ Wu et al. (2018), with various functions can be employed like rectangular Wu et al. (2019), triangular Deng et al. (2022), and linear Wei et al. (2024b). TP-Spikformer employs the triangular-shaped surrogate gradient formulation, described as,

$$\frac{\partial \mathbf{s}_i^\ell[t]}{\partial \tilde{\mathbf{u}}_i^\ell[t]} = \max \left( 0, \beta - |\tilde{\mathbf{u}}_i^\ell[t] - \theta| \right), \tag{23}$$

where $\beta$ is the factor that defines the range of gradient computation, and $\theta$ is the threshold as in Eq. 2. Consequently, the TP-Spikformer can be trained directly with gradient backpropagation.

# L  THEORETICAL ENERGY CONSUMPTION

When analyzing the energy consumption of SNNs, previous studies Yao et al. (2023b); Zhou et al.; 2024a); Wei et al. (2025a;b) commonly assume that MAC and AC operations are implemented on 45nm hardware Horowitz (2014), where $E_{MAC} = 4.6pJ$ and $E_{AC} = 0.9pJ$. To facilitate comparison between different methods, we adopt this approach to theoretically calculate TP-Spikformer's energy consumption, described by the following equation:

$$E_{total} = E_{MAC} \cdot FLOPs_{Conv}^1 + E_{AC} \times (\sum_{n=2}^{N} SOPs_{Conv}^n + \sum_{l=1}^{L} \times SOPs_{Block}^l + SOPs_{MLP}), \tag{24}$$

where $SOPs$ refers to the number of synaptic operations, $SOPs_{conv}^n$ and $SOPs_{MLP}^m$ represent the $SOPs$ for the convolutional operations in the embedding module and the MLP in the classification head, respectively, and $SOPs_{Block}^l$ denotes the $SOPs$ for each transformer block. The number of $SOPs$ in TP-Spikformer is computed as:

$$SOPs^\ell = fr_{Avg}^\ell \times T \times FLOPs^\ell, \tag{25}$$

where $fr_{Avg}$ is the average firing rate of the layer across time steps $T$, and $FLOPs^l$ is the number of floating point operations for the $\ell$-th layer. A spiking transformer typically consists of three components: path embedding, transformer blocks, and the classification head. In TP-Spikformer, the energy consumption of the path embedding and classification head is consistent with the uncompressed counterpart, while the energy consumption of the transformer blocks is significantly reduced.

## M    INSTRUCTIONS FOR USING LARGE LANGUAGE MODELS

In preparing this manuscript, we utilize a large language model (LLM) solely to aid and polish the writing. The LLM is used for grammar checking, language refinement, and improving clarity of expression. It does not contribute to the formulation of research ideas, methodology, experiments, data analysis, or conclusions. All presented in this paper is entirely the work of the authors.

