# OpenReview forum: "TP-Spikformer: Token Pruned Spiking Transformer"
_ICLR.cc/2026/Conference — ICLR 2026 Poster_

### Official Review · Reviewer_BtDR · 2025-10-25

**Soundness:** 3
**Presentation:** 3
**Contribution:** 3
**Rating:** 6
**Confidence:** 4

**Summary:**

This paper presents TP-Spikformer, a method to speed up Spiking Transformers by pruning unimportant tokens. Its main ideas are: 1) a new scoring rule called IRToP, which finds important tokens by checking if they look different from their neighbors (space) and if they change over time (time); and 2) a pruning architecture called IR-Arc, which makes unimportant tokens "skip" the heavy computation in a block instead of deleting them. This keeps the model's structure intact. Experiments show the method works well on many models and tasks, and impressively, it boosts speed with almost no accuracy loss even without any retraining.

**Strengths:**

1. The method is very versatile. It works on simple Spiking Transformers, complex ones with convolutions (like QKFormer), and on many different tasks like classification and detection. This shows it's a useful, general tool.
2. The biggest plus is its "training-free" ability. This is very practical for real-world use where retraining is too expensive.
3. The paper backs up its claims with strong experiments. The ablation studies clearly show why each part of the design is necessary, and the results on big datasets like ImageNet prove that the speedup is real.

**Weaknesses:**

1. The paper doesn't have a good, automatic way to decide how many tokens to prune in each layer. For some models, they just set the numbers by hand. For others, they run a slow and expensive "grid search" beforehand. The reported speedup doesn't include this search time, which makes the method seem faster than it is to set up.
2. The method prunes at the whole "block" level. A token either skips both the Attention and MLP parts, or does both. It might be better to prune them separately (e.g., prune more tokens for Attention and fewer for MLP) to get a better speed-vs-accuracy balance.
3. The paper only considers skipping tokens. Another popular technique is "token merging," where several similar tokens are combined into one. This might preserve more information than just letting old tokens pass through unchanged. The paper should have compared its method to token merging.
4. All experiments were on "directly trained" SNNs. It's unclear if the method would work on ANN-to-SNN conversion.

**Questions:**

1. How long does the grid search for finding the pruning ratios actually take? Have you thought about a simpler or automatic way to find these numbers?
2. Could you explain why you chose to skip tokens instead of merging them?
3. Do you think your method would work for SNNs that are converted from ANNs? For example, you can apply your method to "Masked Spiking Transformer".

---

> ### Author Response · Authors · 2025-11-22
> **Responses to Reviewer BtDR's W1.**
>
> Dear reviewer BtDR, we sincerely appreciate your time and effort in reviewing our manuscript and offering valuable suggestions. In the following, we provide a detailed response to address your concerns one by one.
>
> ## W1: The paper doesn't have a good, automatic way to decide how many tokens to prune in each layer. For some models, they just set the numbers by hand. For others, they run a slow and expensive "grid search" beforehand. The reported speedup doesn't include this search time, which makes the method seem faster than it is to set up.
>
>
> Dear reviewer BtDR, we agree that an automatic way to determine the pruning ratio is important. We conduct an exploration of the automatic pruning rate scheme in our response to your question 1. Here, we specifically address your concerns regarding the grid search.
>
> **The search process is conducted offline, so it does not affect the computational cost or throughput during inference on edge devices. Below, we analyze its time overhead.** The grid search used in our paper is a very simple method, intended to perform a coarse search to identify a reasonable pruning rate. Given a pre-trained model and a global token preservation, its detailed search process is as follows:
>
> - **First is to obtain a set of token preservation ratio combinations.** Specifically, the search space for ratios is restricted to a small set of discrete values, such as 0.9×0.9, 0.8×0.8, 0.75×0.75, 0.6×0.6, etc. Furthermore, we impose a monotonic constraint, requiring the token preservation ratio to decrease progressively from shallow to deeper blocks. This is motivated by the observation that shallow layers capture low-level features and thus require higher token retention, while deeper layers handle high-level semantic information and can tolerate more aggressive token pruning.
>
> - **Second, we randomly sample a small batch of data from the training dataset and evaluate the accuracy for each combination in the combinations set.** The combination with the top-1 accuracy is selected and used for subsequent fine-tuning.
>
> **In our experiments, grid search is only used for SDT-V3.** We summarize the details in the table below, including the discrete search space, the number of candidate combinations, the time to evaluate each combination, and the total search time. **By reviewing the search logs, we observe that different configurations give similar performance under the same global ratio. Therefore, for QKFormer and SDT-V1, we directly set the ratios manually and fine-tune the models, without performing grid search.**
>
> | Model | Global ratio | Searching space per ratio                        | Number of combinations | Evaluation time per combination | Total Time (4 × NVIDIA 4090) |
> |-------|--------------|--------------------------------------------------|------------------------|----------------------------------|------------------------------|
> | SDT-V3 | 0.78         | [1, 0.90, 0.81, 0.72, 0.64, 0.56, 0.49]         | 65                     | 26s                              | 28min 11s                    |
> | SDT-V3 | 0.65         | [0.81, 0.72, 0.64, 0.56, 0.49, 0.42, 0.36]      | 89                     | 22s                              | 32min 38s                    |
> | SDT-V3 | 0.56         | [0.81, 0.72, 0.64, 0.56, 0.49, 0.42, 0.36]      | 166                    | 16s                              | 44min 16s                    |

---

> ### Author Response · Authors · 2025-11-22
> **Responses to Reviewer BtDR's W2, W3, W4.**
>
> ## W2: The method prunes at the whole "block" level. A token either skips both the Attention and MLP parts, or does both. It might be better to prune them separately (e.g., prune more tokens for Attention and fewer for MLP) to get a better speed-vs-accuracy balance.
>
> We agree with you that applying different pruning rates to different modules may yield better results. To explore this view, we have conducted experiments using the SDT-V1-8-768 architecture on the ImageNet dataset, comparing performance under three pruning schemes: **(a) pruning fewer tokens for Attention and more for FFN, (b) pruning more tokens for Attention and fewer for FFN, and (c) applying the same pruning ratio to both modules.** We do not involve fine-tuning for these experiments, leveraging the training-free advantage of TP-Spikformer.
>
> *Experimental Details.* In contrast to the previous method, where token pruning is applied at the start of each transformer block using IRToP, with tokens restored to their feature map at the end of the block, we now apply different pruning ratios to different modules. Specifically, we first perform token pruning before the Attention layer, restoring the feature map after the Attention operation. Tokens are pruned again before the MLP layer, and the feature map is restored after the MLP operation. The table below presents the results of our experiments, where we evaluated three pruning strategies under two ratio settings.
>
> *Experimental Results.* We find that as long as the total pruning rate remains constant, the performance does not change significantly, and that pruning more tokens in the Attention module generally leads to worse results. We hypothesize that combining the adaptive token pruning scheme with automated pruning ratio search could be more effective. Nevertheless, **a potential drawback of this approach is that it increases the number of evaluations required to determine token importance, thereby introducing additional overhead.**
>
> | Pruning Schemes | Attn_ratio | FFN_ratio | Avg_ratio | Accuracy |
> |---|-|--|-----------|----------|
> |(a) | 1.0        | 0.73      | 0.86      | 75.58    |
> |(b) | 0.73       | 1         | 0.86      | 74.39    |
> |(c) | 0.86       | 0.86      | 0.86      | 75.77    |
> |(a) | 0.86       | 0.61      | 0.73      | 74.65    |
> |(b) | 0.61       | 0.86      | 0.73      | 73.47    |
> |(c) | 0.73       | 0.73      | 0.73      | 75.21    |
>
> ---
>
> ## W3: The paper only considers skipping tokens. Another popular technique is "token merging," where several similar tokens are combined into one. This might preserve more information than just letting old tokens pass through unchanged. The paper should have compared its method to token merging.
>
> Thank you for your insightful comment. Token merging has been proposed as a training-free token compression scheme in ANNs, providing an attractive alternative to earlier methods that introduce extra parameters or rely on fine-tuning. **It can be applied to ViT-like spiking Transformer architectures, but it is difficult to extend it to more advanced spiking Transformers such as QKFormer and SDT-V2/V3.** We have explained the reasons for this in detail in our response to your Q2.
>
> Here, **following your suggestion**, we additionally apply the classical token merging method, ToMe[1], to Spikformer and compare it with our TP-Spikformer. As shown in the table below, **our experiments on both datasets demonstrate that TP-Spikformer achieves superior performance with a much lower token retention ratio. This gap would widen further on larger-scale datasets like ImageNet.**
>
> | Method | Architecture| Dataset | Token retention ratio | Accuracy |
> |-------------|------------|--------------|--------|----------|
> |Token Merge| Spikformer-4-384 | CIFAR-10| 0.77| 95.05|
> |Token Merge|Spikformer-4-384|CIFAR-10|0.56| 94.93|
> | **TP-Spikformer** | Spikformer-4-384 | CIFAR-10 | **0.25**                | **95.16**  |
> | **TP-Spikformer** | Spikformer-4-384 | CIFAR-10 | **0.20**                | **95.12**  |
>
> | Method | Architecture| Dataset | Token retention ratio | Accuracy |
> |-------------|------------|--------------|--------|----------|
> |Token Merge|Spikformer-4-384|CIFAR-100|0.77|77.74|
> |Token Merge|Spikformer-4-384|CIFAR-100| 0.56| 77.06|
> | **TP-Spikformer** | Spikformer-4-384 | CIFAR-100 | **0.60**                | **78.48**  |
> | **TP-Spikformer** | Spikformer-4-384 | CIFAR-100 | **0.55**                | **77.83**  |
>
> [1] TOKEN MERGING: YOUR VIT BUT FASTER. ICLR 2023
>
> ---
>
> ## W4: All experiments were on "directly trained" SNNs. It's unclear if the method would work on ANN-to-SNN conversion.
>
> TP-Spikformer is a general method that can be applied to different tasks, architectures, and training paradigms (direct training/conversion-based training). In our response to your Q3, we have added experiments where TP-Spikformer is integrated into a conversion-based algorithm to demonstrate its generality further. We hope this helps to address your concerns.

---

> ### Author Response · Authors · 2025-11-22
> **Responses to Reviewer BtDR's Q1.**
>
> ## Q1: How long does the grid search for finding the pruning ratios actually take? Have you thought about a simpler or automatic way to find these numbers?
>
> Dear reviewer BtDR, we have provided the search results in our response to your W1, including the discrete search space, number of candidate combinations, evaluation time per combination, and total search time. Here, following your suggestion, we conduct a brief exploration of an automatic strategy for determining token pruning ratios.
>
> ### Scheme outline
> In visual tasks, input images can vary significantly; some images have simple backgrounds, while others contain dense and detailed structures. Thus, we design an input-aware adaptive token preservation ratio. Specifically, after computing token importances with IRToP, we define a **polarization coefficient** to measure how unevenly importance scores are distributed. High polarization indicates a few dominant tokens, and an **aggressive pruning strategy** should be applied to retain only these critical tokens. In contrast, low polarization suggests more uniform scores. This usually corresponds to background or transitional regions with no clearly dominant tokens, in which case a **conservative pruning strategy** is preferred to avoid mistakenly removing potentially useful information.
>
> ### Implementation details
>
>
> Given the token importance vector $\mathbf{I}$, we calculate its mean $\mu$ and standard deviation $\sigma$, and then define the coefficient of variation (named CV):
> $$
> \text{CV} = \frac{\sigma}{\mu + \varepsilon},
> $$
> where $\varepsilon$ is a small constant to avoid division by zero. Based on this coefficient, we dynamically assign the token retention ratio using the following piecewise linear function:
> $$
> \text{ratio} _ {\text{keep}} =
> \begin{cases}
> \operatorname{clip}\big(0.5 - 0.3 \cdot (\text{CV} - 1.0),\, 0.2,\, 0.5\big), & \text{if } \text{CV} > \theta, \\
> \operatorname{clip}\big(0.5 + 0.3 \cdot (1.0 - \text{CV}),\, 0.5,\, 0.8\big), & \text{if } \text{CV} \le \theta.
> \end{cases}
> $$
> We set the threshold $\theta=1$ to divide input features into two cases. **If $\text{CV}>\theta$, token importance varies widely, so we prune more aggressively but keep at least 20% of tokens. If $\text{CV}≤\theta$, token importance is more uniform, so we keep more tokens but no more than 80% to avoid redundancy.** Finally, the number of tokens to keep is $k = \lfloor \text{ratio} _ {\text{keep}} \cdot N _ v \rfloor$.
>
>
> ### Experiment and summary
>
> We integrate this scheme into TP-Spikformer and evaluate it on SDT-V1. Without fine-tuning, it achieves 72.08% accuracy with an average token preservation ratio of 0.41. After 50 epochs of fine-tuning, the accuracy improves to 73.71%, with the ratio still fixed at 0.41. Through this experiment, we aim to demonstrate the viability of input-aware adaptive pruning. **Maybe this scheme is still not perfect, we hope this preliminary exploration during the limited rebuttal period helps addresses your concern.**

---

> ### Author Response · Authors · 2025-11-22
> **Responses to Reviewer BtDR's Q2, Q3.**
>
> ## Q2: Could you explain why you chose to skip tokens instead of merging them?
>
> Yes, certainly. Before explaining this, we first clarify the additional challenges that token pruning in SNNs faces compared with its counterpart in ANNs.
>
> The key difference between Spiking Transformers and ANN Transformers is that **the former interleave Transformer blocks with convolutional layers**. Specifically, early SNN transformers like Spikformer and SDT-V1, follow ViT-style designs from ANNs, using patch embedding and standard transformer blocks. As the field developed, recent SOTA models like QKFormer and SDT-V2/V3 incorporate convolution layers inside transformer blocks. For example, QKFormer applies conv-based Spiking Patch Embedding before each block, and SDT-V3 uses spike-based separable convolutions before every attention layer.  **Unlike ANNs where features can be flattened for token removal, these convolutional layers embedded in the transformer blocks present new challenges for token compression in SNNs,** which can be understood from two aspects:
> - On the one hand, convolution operations rely on structured grid-like inputs. This means that **if tokens are removed from a transformer block, the remaining tokens may no longer form a valid image layout, making them incompatible with later convolutional layers**.
> - On the other hand, due to its strong prior assumptions, e.g., spatial local correlation and translation invariance, the convolution operation heavily relies on the spatial structure of feature maps. **However, removing tokens disrupts the spatial structure of feature maps. This disruption (1) impairs local information propagation, (2) degrades the effectiveness of trained filters, and (3) compromises the model's representational ability.**
>
> Thus, **when performing token pruning on SOTA Spiking Transformers, it is crucial to preserve the completeness of the feature maps and their spatial structure.**
>
>
> In ANNs, token merge adopts a graph-theoretic view that **treats the feature map as a 1D sequence of tokens.** It partitions all tokens into two disjoint sets ($src$ and $dst$). For each token in src, it computes its similarity to all tokens in dst, thus constructing a weighted bipartite graph between the two sets based on these similarities. Then, it may apply a threshold to keep only high-similarity edges, yielding a subgraph. Finally, it uses the node degree in this subgraph as a redundancy measure and merge tokens with higher node degree. **This process inherently breaks the regular 2D grid structure.** Because convolution layers in Spiking Transformers strongly rely on a regular spatial grid, such graph-based **token merging cannot be directly applied to token pruning in SNNs.** In contrast, **we place the updated informative tokens and the uninformative tokens back in their original spatial locations after block computation, reconstructing a complete feature map**. This explains why skipping tokens is feasible, whereas merging them is not.
>
> -----------------------------------------------------------------------------
>
> ## Q3: Do you think your method would work for SNNs that are converted from ANNs? For example, you can apply your method to "Masked Spiking Transformer".
>
> Yes. TP-Spikformer can also be applied to conversion-based training. To verify this, we integrate it into the classical *Masked Spiking Transformer* and evaluate it on CIFAR-10 under two pruning ratios. As shown in the Table below, with 256 time steps, our method achieves 95.11% and 96.04% accuracy at token preservation ratios of 0.29 and 0.57, respectively. The original ANN model reaches 96.72% accuracy before conversion. The performance gap before and after conversion falls within the inevitable error introduced by the conversion process. **These results indicate that TP-Spikformer can be effectively extended to conversion-based training frameworks.** We hope this helps to address your concerns. Moreover, we will add the corresponding discussion in our revised manuscript.
>
> | Architecture| Dataset | # Param|Token retention ratio |Time steps| Accuracy |
> |:----:|:----:|:----:|:----:|:----:|:----:|
> |Swin-T (BN)| CIFAR-10|27.6 M|0.57|64|95.07|
> |Swin-T (BN)|CIFAR-10|27.6 M|0.57|128|95.80|
> |Swin-T (BN)|CIFAR-10|27.6 M|0.57|256|96.04|
>
> | Architecture| Dataset | # Param|Token retention ratio |Time steps| Accuracy |
> |:----:|:----:|:----:|:----:|:----:|:----:|
> |Swin-T (BN)|CIFAR-10|27.6 M|0.29|64|94.47|
> |Swin-T (BN)|CIFAR-10|27.6 M|0.29|128|94.98|
> |Swin-T (BN)|CIFAR-10|27.6 M|0.29|256|95.11|

---

### Official Review · Reviewer_G33Y · 2025-10-29

**Soundness:** 3
**Presentation:** 3
**Contribution:** 2
**Rating:** 4
**Confidence:** 5

**Summary:**

This paper proposed an information retaining token pruning framework for spiking transformers.

**Strengths:**

1. This paper proposed an information retaining token pruning framework for spiking transformers.

2. The writing in this paper is good.

3. The experiments are quite extensive, including classification, segmentation, detection, and tracking tasks.

**Weaknesses:**

1. My main concern lies in the motivation. Currently, directly trained spiking Transformers are relatively small- or medium-scale models. Although pruning slightly reduces performance while improving throughput（eg, TP-Spikformer with SDT-V1-8-768 on imagenet: -1.53% acc, thr 29%）， this trade-off does not constitute a strong motivation.

2、There is a lack of discussion on overall model training costs, such as training time and memory consumption.

3、There is a lack of spike-driven characteristics and energy consumption discussion on the heuristic spatiotemporal information-retaining criterion.

**Questions:**

see above

---

> ### Author Response · Authors · 2025-11-20
> **Responses to Reviewer G33Y's W1, W2.**
>
> Dear Reviewer G33Y, we sincerely appreciate your time and effort in reviewing our manuscript and offering valuable suggestions. In the following, we provide a detailed response to address your concerns one by one.
>
>
> ## W1: My main concern lies in the motivation. Currently, directly trained spiking Transformers are relatively small- or medium-scale models. Although pruning slightly reduces performance while improving throughput（eg, TP-Spikformer with SDT-V1-8-768 on imagenet: -1.53% acc, thr 29%）, this trade-off does not constitute a strong motivation.
>
> Dear reviewer, thank you for your valuable comment. The “small-/medium-scale models” you mentioned mainly refer to **parameter count or model size**. From this viewpoint, further weight quantization or pruning may indeed yield limited benefits. However, TP-Spikformer does not focus on weight compression; instead, it targets **token pruning in intermediate feature maps**, which is crucial regardless of model scale.
>
> We explain it below in detail. In Spiking Transformers, intermediate feature maps are often large. For example, in SDT-V1-8-768, the feature map of a single layer at one time step is of size $14 \times 14 \times 768 $. Such feature maps typically cannot be fully buffered in on-chip SRAM and must be stored in off-chip memory (e.g., DDR/DRAM), with frequent transfers between on-chip and off-chip memory. Thus, **the number of tokens directly determines the frequency of off-chip accesses.** Unfortunately, as per Fig3.3 in paper [1], off-chip DRAM access (640pJ) requires 128× more energy than on-chip SRAM access (5pJ) and 6400× more than basic integer addition (0.1pJ), so **off-chip access is usually the dominant source of latency and energy consumption in deployment.** Luckily, TP-Spikformer directly reduces the token number, thereby decreasing the off-chip memory access.
>
>
> **In short, independent of model scale, once deploy Spiking Transformer on resource-limited neuromorphic hardware, pruning tokens in intermediate feature maps to reduce off-chip access is both meaningful and practically necessary. This is the core motivation of TP-Spikformer, rather than merely pursuing throughput gains on general-purpose GPUs.**
>
> > [1] Efficient processing of deep neural networks. 2020.
>
> -----------------------------------------------------------------------------
>
> ## W2: There is a lack of discussion on overall model training costs, such as training time and memory consumption.
>
> Dear reviewer G33Y, thank you for your insight comments. Although our method is primarily designed to optimize inference efficiency, it can also contribute to efficiency gains during training. **As per your suggestion, we conduct experiments on large-scale ImageNet using SDT-V3-5M with and without our token pruning method, both trained from scratch.** Both models are trained on a single H800 GPU, following the original SDT-V3 settings to ensure consistency and fair comparison. The results are shown in the table below. **TP-Spikformer achieves a top-1 accuracy of 73.6%, which is comparable to the 73.9% of the unpruned SDT-V3, while reducing training time by 7.5 hours.** This demonstrates the acceleration effect during training. **Moreover, due to fewer tokens, TP-Spikformer uses significantly less GPU memory, allowing larger models or batch sizes to be trained on the same hardware.** These results confirm that our TP-Spikformer not only offers inference efficiency but also acceleration advantages during training, making it especially useful in resource-constrained scenarios.
>
> | Model         | Params (M) | Token preservation ratio | Total training time | GPU Memory (batch size = 1280) | Acc.   |
> |--------------|-----------|--------------------------|---------------------|---------------------------------|--------|
> | SDT-V3       | 5.1        | 1.00                     | 57h 14min           | 77.49 GB                        | 73.9%  |
> | TP-Spikformer| 5.1        | 0.65                     | 50h 49min           | 66.90 GB                        | 73.6%  |

---

> ### Author Response · Authors · 2025-11-20
> **Responses to Reviewer G33Y's W3.**
>
> ## W3: There is a lack of spike-driven characteristics and energy consumption discussion on the heuristic spatiotemporal information-retaining criterion.
>
>
>
> Thank you for raising this important issue. As per your suggestion, we add a discussion of the spike-driven characteristics and energy consumption of the IRToP.
>
> **First, spike-driven characteristics.** The IRToP criterion computes token similarity and perform normalization, which may involve operations that are not fully hardware-friendly. When designing TP-Spikformer, we carefully considered this issue and its potential drawbacks. After thorough analysis, we choose to retain these computation for the two reasons:
>
> - *Performance Considerations.* Because spike-driven computation relies on discrete signals, **a fully spike-driven scorer may struggle to provide the fine-grained importance estimates needed for effective token pruning. This precision is especially critical for token selection in sparsely operated, tightly coupled convolution-transformer structure in exisiting SOTA Spiking Transformer (such as QKFormer and SDT-V2/V3).** If we force the scorer to be purely spike-driven, it would make pruning decisions less reliable and likely cause a noticeable accuracy drop. Thus, using non-spike-driven scoring modules is a common design choice in SNNs [1,2].
>
> - *Hardware Feasibility.* The IRToP only needs to be implemented once in hardware and can then be reused. While  it involves some operations that are not inherently hardware-friendly, many recent works propose neuromorphic-friendly or low-cost approximations to this computation [3,4,5,6]. **The increasing maturity of hardware technology provides support for this approach. Moreover, we provide an detailed energy evaluation of IRToP in the following, which shows that the additional cost introduced by IRToP is very small.**
>
> In summary, the IRToP scorer is designed with careful consideration of balancing high-performance
> requirements, the availability of mature hardware support, and the energy efficiency.
>
> [1] AT-SNN: Adaptive Tokens for Vision Transformer on Spiking Neural Network. Arxiv 2024.
>
> [2] Towards efficient spiking transformer: a token sparsification framework for training and inference acceleration. ICML 2024.
>
> [3] Comparison of semantic vectors with reduced precision using the cosine similarity measure. IEEE IntelliSys.
>
> [4] COSIME: FeFET based Associative Memory for In-Memory Cosine Similarity Search. IEEE/ACM International Conference on Computer-Aided Design.
>
> [5] Softermax: Hardware/software co-design of an efficient softmax for transformers. DAC 2021.
>
> [6] Hardware-Efficient SoftMax Architecture With Bit-Wise Exponentiation and Reciprocal Calculation. IEEE Transactions on Circuits and Systems I: Regular Papers 2024.
>
> **Second, energy consumption.** Following your suggestion, we recompute the energy consumption of TP-Spikformer, showing that IRToP introduces only a small additional energy overhead. Let the input to layer $\ell$ be $\mathbf{X}^{\ell-1} \in \mathbb{R}^{T \times H \times W \times D}$ At time $t$, there are $H \times W$ tokens, each of dimension $D$. The spatial scorer computes one cosine similarity per token, followed by a softmax normalization over the resulting $H \times W$ spatial scores. The temporal scorer computes, for each token, its difference from the corresponding token at time $t-1$ without multiplications, followed by another normalization over $H \times W$ temporal scores. Finally, the spatial and temporal scores are summed elementwise. Therefore, the multiplications and additions of IRToP at time $t$ in layer $\ell$ are given by
> $$
> \textnormal{Mul} _ {\textnormal{IRToP}} \approx H \times W \times \textnormal{Mul} _ {\textnormal{cosine}} + 2 \times \textnormal{Mul} _ {\textnormal{softmax}},
> $$
> and
> $$
> \textnormal{Add} _ {\textnormal{IRToP}} \approx H \times W \times \textnormal{Add} _ {\textnormal{cosine}} + D \times H \times W + 2 \times \textnormal{Add} _ {\textnormal{softmax}} + H \times W.
> $$
>
>
>
> Despite several hardware-efficient implementations of cosine similarity and softmax normalization, we use their dense versions for our energy estimation. *Their required numbers of multiplications and additions are given in Theorems 1 and 2.* **Based on this, we recalculate the energy of SDT-V1-8-768 on Imagenet-1K, as shown below. Clearly, the proposed scorer introduces only a small additional energy overhead.**
>
> | Model| Time | Token Preservation Ratio |OPs$ _ {blocks}$ (G)|Power (mJ)|Accuracy (%)|Throughput (imgs/s)   |
> | --- | :--: | :---: | :---: | :---: | :----: | :-----: |
> | SDT-V1-8-768  |  4   |×1|    9.04 (Base)|    10.26 (Base)|   76.32 (Base)|   156 (Base)|
> | TP-Spikformer |  4   |×0.74| 6.78 (↓25.0%)| 8.27 (↓19.4%)| 75.82 (−0.50) | 181 (↑16.0%)|
> | TP-Spikformer |  4   |×0.65| 5.96 (↓34.1%)| 7.53 (↓26.6%)| 75.62 (−0.70) | 189 (↑21.2%)|
> | TP-Spikformer |  4   |×0.51| 4.74 (↓47.6%)| 6.43 (↓37.3%)| 74.79 (−1.53) | 202 (↑29.5%)|

---

> ### Author Response · Authors · 2025-11-20
> **Theorem 1**
>
> **Theorem 1 (Multiplications and Additions of cosine similarity between two $D$-dimensional vectors).**
> Let $\mathbf{x}, \mathbf{y} \in \mathbb{R}^D$ and $\textnormal{cosine}(\mathbf{x}, \mathbf{y}) = \frac{\mathbf{x}^\top \mathbf{y}}{\|\mathbf{x}\| _ 2 \|\mathbf{y}\| _ 2}$. Assume that the dot product and squared $\ell_2$-norms are computed by straightforward accumulation, i.e., $\mathbf{x}^\top \mathbf{y} = \sum _ {k=1}^{D} x _ k y _ k$, $\|\mathbf{x}\| _ 2^2 = \sum _ {k=1}^{D} x _ k^2$, and $\|\mathbf{y}\| _ 2^2 = \sum _ {k=1}^{D} y _ k^2$. Ignoring the constant cost of the scalar square-root and division operations in the final normalization, the total numbers of scalar multiplications and additions required to compute $\textnormal{cosine}(\mathbf{x}, \mathbf{y})$ are approximated as $\textnormal{Mul} _ {\textnormal{cosine}} \approx 3D$ and $\textnormal{Add} _ {\textnormal{cosine}} \approx 3D - 3$.
>
>
>
> **Proof.**
> The computation is decomposed into three stages: **(i) dot-product evaluation, (ii) squared-norm computation, and (iii) normalization.**
>
> 1. **Dot product.**
>    The dot product $\mathbf{x}^\top \mathbf{y} = \sum _ {k=1}^{D} x _ k y _ k$ is computed by multiplying and accumulating elementwise products. This requires $D$ multiplications and $D-1$ additions:
>    $$
>    \textnormal{Mul} _ {\textnormal{dot}} = D, \qquad
>    \textnormal{Add} _ {\textnormal{dot}} = D - 1.
>    $$
>
> 2. **Squared norms.**
>    The squared norms are given by $\|\mathbf{x}\| _ 2^2 = \sum _ {k=1}^{D} x _ k^2$ and $\|\mathbf{y}\| _ 2^2 = \sum _ {k=1}^{D} y _ k^2$. Each squared norm uses $D$ multiplications and $D-1$ additions, so together they require
>    $$
>    \textnormal{Mul} _ {\textnormal{norm}^2} = 2D, \qquad
>    \textnormal{Add} _ {\textnormal{norm}^2} = 2D - 2.
>    $$
>
> 3. **Normalization.**
>    The cosine similarity is obtained as $\textnormal{cosine}(\mathbf{x}, \mathbf{y}) = \mathbf{x}^\top \mathbf{y} / (\|\mathbf{x}\| _ 2 \|\mathbf{y}\| _ 2)$, which involves two scalar square-root operations and one scalar division (and possibly one scalar multiplication for forming the denominator). The one-time cost of these scalar operations is $O(1)$, is absorbed into the constant factor, and is thus omitted from the MAC/AC counts above.
>
> Summing the contributions of these two stages yields
> $$
> \begin{aligned}
> \textnormal{Mul} _ {\textnormal{cosine}}
> = \textnormal{Mul} _ {\textnormal{dot}} + \textnormal{Mul} _ {\textnormal{norm}^2}
> = D + 2D = 3D,
> \end{aligned}
> $$
> and
> $$
> \begin{aligned}
> \textnormal{Add} _ {\textnormal{cosine}}
> = \textnormal{Add} _ {\textnormal{dot}} + \textnormal{Add} _ {\textnormal{norm}^2}
> = (D - 1) + (2D - 2) = 3D - 3.
> \end{aligned}
> $$
>
> These expressions characterize the arithmetic complexity of computing the cosine similarity. $\square$

---

> ### Author Response · Authors · 2025-11-20
> **Theorem 2**
>
> **Theorem 2 (Multiplications and Additions of a length-$N$ Softmax with polynomial exponential).**
> Let $\mathbf{z} \in \mathbb{R}^N$ and $\operatorname{softmax}(z _ i) = \frac{\exp(z _ i)}{\sum _ {j=1}^{N}\exp(z _ j)}, i = 1,\dots,N$. Typically, $\exp(t)$ is implemented in hardware by an $x$-term polynomial, i.e., $\exp(t) \approx a _ 0 + a _ 1 t + \dots + a _ {x-1} t^{x-1}$, evaluated using Horner’s rule ($x = 4$ usually provides sufficient approximation accuracy). Under these assumptions and neglecting the constant cost of computing the scalar reciprocal $1/S$, the total numbers of scalar multiplications and additions required to compute the Softmax of $\mathbf{z}$ are approximated as $\textnormal{Mul} _ {\textnormal{softmax}} \approx Nx$, $\textnormal{Add} _ {\textnormal{softmax}} \approx Nx - 1$.
>
>
>
> **Proof.**
> The computation can be decomposed into three stages: **(i) evaluation of the exponential approximation, (ii) computation of the denominator, and (iii) normalization.**
>
> 1. *Exponential approximation.*
>    For a single input $t = z _ i$, Horner’s rule evaluates the $x$-term polynomial via the recurrence $y \leftarrow y \cdot t + a _ k, \quad k = x-2,\dots,0,$ which executes exactly $x-1$ iterations. Each iteration performs one multiplication and one addition. Hence, for one element,
>    $$
>    \textnormal{Mul} _ {\exp,1} = x - 1, \qquad
>    \textnormal{Add} _ {\exp,1} = x - 1.
>    $$
>    Since the same procedure is applied independently to all $N$ entries of $\mathbf{z}$, the total cost of exponential evaluation is
>    $$
>    \textnormal{Mul} _ {\exp} = N(x - 1), \qquad
>    \textnormal{Add} _ {\exp} = N(x - 1).
>    $$
>
> 2. *Denominator computation.*
>    Let $e _ i = \exp(z _ i)$ denote the approximated exponentials. The denominator $S = \sum _ {i=1}^{N} e _ i$ can be computed by a simple accumulation, which requires exactly $N-1$ additions and no multiplications:
>    $$
>    \textnormal{Mul} _ {\textnormal{sum}} = 0, \qquad
>    \textnormal{Add} _ {\textnormal{sum}} = N - 1.
>    $$
>
> 3. *Normalization.*
>    For the single reciprocal $r = 1/S$, whose one-time computation cost is $O(1)$ and absorbed into the constant factor and thus omitted. Each output is given by $\hat{y} _ i = e _ i \cdot r, \quad i = 1,\dots,N.$ This step uses one multiplication per element and no additions, i.e.,
>    $$
>    \textnormal{Mul} _ {\textnormal{norm}} = N, \qquad
>    \textnormal{Add} _ {\textnormal{norm}} = 0.
>    $$
>
> Summing the contributions of all three stages yields
>
> $$
> \begin{aligned}
> \textnormal{Mul} _ {\textnormal{softmax}}= \textnormal{Mul} _ {\exp} + \textnormal{Mul} _ {\textnormal{sum}} + \textnormal{Mul} _ {\textnormal{norm}} = N(x - 1) + 0 + N = Nx,
> \end{aligned}
> $$
> and
> $$
> \begin{aligned}
> \textnormal{Add} _ {\textnormal{softmax}}
> = \textnormal{Add} _ {\exp} + \textnormal{Add} _ {\textnormal{sum}} + \textnormal{Add} _ {\textnormal{norm}}
> = N(x - 1) + (N - 1) + 0
> = Nx - 1.
> \end{aligned}
> $$
>
> This establishes the claimed expressions for the arithmetic complexity. $\square$

---

> ### Author Response · Authors · 2025-11-27
>
> Dear Reviewer G33Y,
>
> We sincerely appreciate your time and effort in reviewing our manuscript and offering valuable suggestions. As the author-reviewer discussion phase is drawing to a close, we would like to confirm whether our responses have effectively addressed your concerns. We have provided detailed responses to your concerns, and we hope they have adequately addressed your issues. If you require further clarification or have any additional concerns, please do not hesitate to contact us. We are more than willing to continue our communication with you.
>
> Best regards.

---

### Official Review · Reviewer_r4rD · 2025-10-30

**Soundness:** 3
**Presentation:** 3
**Contribution:** 3
**Rating:** 6
**Confidence:** 5

**Summary:**

This paper introduces TP-Spikformer, a simple and effective token pruning method designed to reduce the high computational and storage costs of large-scale spiking transformers, making them more suitable for resource-constrained devices. The core of the method consists of two main contributions. First, it proposes a heuristic Information-Retaining Token Pruning (IRToP) criterion, which scores and identifies important tokens by evaluating both their spatial distinctiveness from neighbors and their temporal variation across time steps . Second, it introduces the Information-Retaining Architecture (IR-Arc), which, instead of permanently dropping unimportant tokens, applies a block-level early stopping strategy—uninformative tokens bypass computation within a block and are then reassembled with the processed informative tokens. A significant advantage of this approach is its versatility and efficiency, as it requires no retraining or architectural modifications, achieving competitive performance even with zero fine-tuning. The authors demonstrate the method's effectiveness, efficiency, and scalability across diverse architectures (like Spikformer, QKFormer, and SDT) and a range of visual tasks, including classification, detection, segmentation, and tracking.

**Strengths:**

This paper presents a robust and highly impactful contribution, demonstrating exceptional strengths across originality, quality, clarity, and significance. The work's originality is outstanding, introducing a novel, training-free token pruning framework for spiking transformers, which stands in stark contrast to prior methods that require costly retraining and architectural modifications. The core ideas are creative and well-motivated: the bio-inspired IRToP criterion offers a new heuristic for identifying important tokens based on both spatial saliency and temporal dynamics, while the IR-Arc architecture cleverly uses a block-level early stopping and reassembly strategy instead of direct token removal, making it compatible with modern hierarchical SNNs. The research quality is superb, validated through extensive experiments across a diverse set of architectures (Spikformer, SDT-V1/V3, QKFormer) and a wide range of tasks, including classification, detection, segmentation, and tracking. The results consistently show significant reductions in computational overhead with minimal performance degradation, and thorough ablation studies rigorously justify the design choices. The paper is presented with excellent clarity, logically progressing from a well-defined problem to an elegant solution, with effective visualizations and a clear algorithm to aid understanding. Finally, the work is of high significance as it provides a simple, practical, and broadly applicable tool that addresses a critical bottleneck in deploying large-scale SNNs. By enabling efficient compression without retraining, TP-Spikformer dramatically lowers the barrier for applying spiking transformers in resource-constrained, real-world scenarios, marking a key step forward for the field of energy-efficient neuromorphic computing.

**Weaknesses:**

1. The paper would be strengthened by providing further experimental results, such as from an entropy or visualization perspective, to more rigorously justify the effectiveness of the Spatial and Temporal token scorers in measuring token importance.
2. The authors state that TP-Spikformer performs well even without training, but its accuracy drops significantly when using QKFormer and SDT-V3.

**Questions:**

1. Can the proposed method be applied to train a spiking transformer from scratch?
2. How could the proposed method be generalized to non-grid-like modalities?
3. How well does the proposed method preserve key information during pruning?What are the key differences between token pruning in spiking transformers and conventional ANN-based transformers?

---

> ### Author Response · Authors · 2025-11-20
> **Responses to Reviewer r4rD's W1, W2.**
>
> Dear reviewer r4rD, thank you for your professional and insightful comments on our paper. In the following, we address your questions one by one.
>
> ## W1: The paper would be strengthened by providing further experimental results, such as from an entropy or visualization perspective, to more rigorously justify the effectiveness of the Spatial and Temporal token scorers in measuring token importance.
>
> As per your suggestion, we visualize the spatial and temporal scores to better understand their roles.  We use the SDT-V1-8-768 architecture and display token scores from the last block before the classification head. The visualization is provided at the anonymous link: https://anonymous.4open.science/r/TP-Spikformer-Token-Pruned-Spiking-Transformer-6963/score.pdf, where we obtain two observations:
>
> - The spatial scorer assigns higher keep scores to tokens that are spatially distinctive from their surroundings. In natural images, this saliency often appears along **edges and textures**. As shown in the figure, **it tends to emphasize the bird’s contour, which is consistent with our local-contrast-based design in spatial scorer.**
>
> - The temporal scorer assigns higher keep scores to tokens with stronger temporal variation. Due to the nonlinear dynamics of spiking neurons, it is difficult to precisely characterize the temporal information of static images encoded in hidden layers. Nevertheless, **the visualization shows that the temporal scorer tends to highlight key parts of the bird (e.g., claws, wings, beak)**, indicating that it can capture semantically discriminative regions.
>
> -----------------------------------------------------------------------------
>
> ## W2: The authors state that TP-Spikformer performs well even without training, but its accuracy drops significantly when using QKFormer and SDT-V3.
>
> We apologize for the unclear expression. The claim that "TP-Spikformer can also perform well in a training-free manner" is made in a **comparative** sense. Existing advanced token pruning methods in SNNs often modify the original model architecture when applied to spiking transformers. These modifications may include introducing new tokens (STATA[1]), adding trainable modules (ACT[2]). **Since these additions are randomly initialized, they require full retraining, which significantly increases data requirements, training costs, and reduces generalizability. Therefore, though our method does not completely preserve accuracy, it achieves better accuracy than existing spiking token pruning methods under the same no-fine-tuning setting.**
>
> We then explain the lower performance of QKFormer and SDT-V3. A key difference between the SOTA spiking transformers (QKFormer, SDT-V3) and SDT-V1 is the frequent use of convolution embedded in blocks. For example, QKFormer applies conv-based Spiking Patch Embedding before each block, and SDT-V3 uses spike-based separable convolutions before every attention layer. Therefore, SDT-V1 maintains high accuracy since its relatively simple architecture, while SOTA spiking transformers suffer from greater accuracy drops due to the inclusion of structure-sensitive convolutional modules. **Overall, the convolution embedded within SOTA transformer makes it difficult to directly apply existing token pruning methods to them, and TP-Spikformer is the first method successfully applied to such complex architectures without relying on training from scratch.**
>
> > [1] Towards efficient spiking transformer: a token sparsification framework for training and inference acceleration. ICML 2024.
>
> > [2] AT-SNN: Adaptive Tokens for Vision Transformer on Spiking Neural Network. Arxiv 2024.

---

> ### Author Response · Authors · 2025-11-20
> **Responses to Reviewer r4rD's Q1, Q2.**
>
> ## Q1: Can the proposed method be applied to train a spiking transformer from scratch?
>
> Yes, although our method is mainly designed to improve deployment efficiency, it can also be easily applied during training. To verify this, **we conduct experiments on large-scale ImageNet using SDT-V3-5M with and without our token pruning method, both trained from scratch.** All experiments are run on a single H800 GPU, using the same settings as the original SDT-V3 to ensure a fair comparison. As shown below, **TP-Spikformer achieves 73.6% accuracy, close to the 73.9% of the unpruned model, while reducing training time by 7.5 hours.** This shows its effectiveness in speeding up training. **Moreover, TP-Spikformer uses much less GPU memory due to fewer tokens, which allows larger models or batch sizes to be trained on the same hardware.**
>
> | Model| Params (M) | Token preservation ratio | Total training time | GPU Memory (batch size = 1280) | Acc.   |
> |----|----|----|--|-|----|
> | SDT-V3| 5.1| 1.00| 57h 14min| 77.49 GB| 73.9%  |
> | TP-Spikformer| 5.1| 0.65| 50h 49min| 66.90 GB| 73.6%  |
>
> ---
>
> ## Q2: How could the proposed method be generalized to non-grid-like modalities?
>
> As per your suggestion, we evaluate TP-Spikformer on an NLP task using SpikeLM [1]. Experiments are conducted on the GLUE benchmark with a 12-layer BERT-based encoder, using a token preservation ratio of [1, 1, 1, 0.9, 0.9, 0.9, 0.8, 0.8, 0.8, 0.7, 0.7, 0.7], while keeping all other settings unchanged. As shown below, **TP-Spikformer achieves a performance of 75.9%, showing no significant loss. These results confirm the effectiveness of our method on NLP tasks, demonstrating its general applicability beyond the vision domain.**
>
> | Model| SST-2 | MRPC | RTE  | MNLI$_m$ | QNLI | QQP$_{F1}$  | CoLA | STS-B | Avg. |
> |--|:---:|:--:|:--:|:---:|:--:|:--:|:--:|:---:|:--:|
> | SpikeLM| 87.0  | 85.7 | 69.0 | 77.1 | 85.3 | 83.9 | 38.8 | 84.9  | 76.5 |
> | TP-Spikformer| 87.9  | 84.7 | 68.2 | 76.0 | 84.6 | 84.2 | 37.0 | 84.9  | 75.9 |
>
>
> [1] SpikeLM: Towards General Spike-Driven Language Modeling via Elastic Bi-Spiking Mechanisms. ICML2024.

---

> ### Author Response · Authors · 2025-11-20
> **Responses to Reviewer r4rD's Q3.**
>
> ## Q3: How well does the proposed method preserve key information during pruning? What are the key differences between token pruning in spiking transformers and conventional ANN-based transformers?
>
> Thank you for your professional review. We address your concerns as follows.
>
> *How well does the proposed method preserve key information during pruning?* Under resource constraints, we should recognize that information loss from token pruning is inevitable. **In such cases, the goal of TP-Spikformer is not to preserve all information perfectly (which is often infeasible), but to prioritize retaining the most important information.** Our IRToP method achieves this by discarding low-information tokens, enabling the model to allocate limited computational resources to more informative ones in the image. This is the meaning of "optimally" in our sentences. Therefore, although IRToP does not ensure lossless feature preservation, it maximizes the feature extraction ability of TP-Spikformer by reducing the impact of pruning on critical representations, as supported by our experimental results.
>
> *What are the key differences between token pruning in spiking transformers and conventional ANN-based transformers?* **The key distinction is that Spiking Transformers interleave Transformer blocks with convolutional layers**. Specifically, early SNN transformers, such as Spikformer and SDT-V1, follow ViT-style designs from ANNs, using patch embedding and standard transformer blocks. As the field developed, recent SOTA models like QKFormer and SDT-V2/V3 incorporate convolution layers with kernels larger than 1 inside transformer blocks. For example, QKFormer applies conv-based Spiking Patch Embedding before each block, and SDT-V3 uses spike-based separable convolutions before every attention layer. **Unlike ANNs where features can be flattened for token removal, these convolutional layers embedded in the transformer blocks present new challenges for token compression in SNNs,** which can be understood from two aspects:
> - On the one hand, convolution operations rely on structured grid-like inputs. This means that **if tokens are removed from a transformer block, the remaining tokens may no longer form a valid image layout, making them incompatible with later convolutional layers**.
> - On the other hand, due to its strong prior assumptions, e.g., spatial local correlation and translation invariance, the convolution operation heavily relies on the spatial structure of feature maps. **However, removing tokens disrupts the spatial structure of feature maps. This disruption (1) impairs local information propagation, (2) degrades the effectiveness of trained filters, and (3) compromises the model's representational ability.**
>
> Therefore, when pruning advanced spiking transformers like QKFormer and SDT-V2/V3, it is essential to preserve the overall architectural integrity. Notably, existing SNN token pruning methods have only been tested on ViT-like Spikformer and spike-driven transformer V1, and have not yet been applied to recent SOTA spiking transformers. To the best of our knowledge, we are the first to evaluate token pruning on these advanced spiking architectures.

---

> ### Comment · Reviewer_r4rD · 2025-11-24
> **Response to the Rebuttal**
>
> Thank you for the authors’ thorough reply; it has addressed my questions. After taking the rebuttal into account, I think the paper is well-motivated and experimentally sound, so I will be increasing my score.

---

> > ### Author Response · Authors · 2025-11-25
> >
> > Dear reviewer r4rD, we sincerely appreciate your recognition of our work. We will thoroughly revise the manuscript in accordance with your suggestions and those of the other reviewers to enhance its quality.

---

### Official Review · Reviewer_TA1d · 2025-10-31

**Soundness:** 3
**Presentation:** 3
**Contribution:** 1
**Rating:** 2
**Confidence:** 5

**Summary:**

This paper introduces a token pruning method for spiking transformers. First, a heuristic spatio-temporal information-retaining criterion is proposed to evaluate token importance. Second, based on the importance scores, a block-level early stopping strategy is proposed for uninformative tokens. Experiments are conducted on Spikformer, QKFormer and Spike-driven Transformer V1 and V3.

**Strengths:**

1. The narrative is very convincing that token-level sparsification is a very effective  way for transformer-style models to boost computational  efficiency.

2. This paper's experiments are very solid and extensive. The token sparsification method is verified across different tasks and multiple datasets, which makes the effectiveness very convincing.

3. This work demonstrates the most advanced token  pruning results for spiking ViT in comparison with previous spiking token  pruning methods.

**Weaknesses:**

The substantive contributions of this paper significantly overlap with previous work on ANN transformer sparsification, demonstrating limited novelty. It essentially replicates the success of existing ANN transformer sparsification approaches, with even the narrative framework bearing striking resemblances.

1. The spatial token scorer is highly similar to the ANN token-pruning one [1]. Thus, the novelty of this paper is significantly challenged.

2. The ablation study in Table 6 is conducted on ImageNet, a static dataset with no temporal disparity, thus I cannot confirm the effectiveness of the temporal pruning scorer. The visualization in Figure 5 does not entirely prove the effectiveness of the temporal pruning scorer. Instead, it exhibits limitations in spatial perception of the spatial scorer.

3. It is very counterintuitive that the temporal scorer is used to prune spatial token rather than prune temporal token. Moreover, temporal pruning is applied regardless of whether the dataset is dynamic or static.

4. The retaining is already existent. This is also called token emerger technique. Or, retaining can be seen as a naive token emerger that can restore non-informative tokens for future use. I can easily give an example of such retaining [2].

In short, this paper shows severe duplication of ANN token-pruning domain. In essense, the success of this paper's method is inherited from the ANN  token-pruning  works. Such that this paper's contribution is limited. Moreover, the spatial token pruning may be even better if the most advanced ANN token pruning is applied.

This is more like a very good technique report concerning the verification of ANN methods applied to SNNs.

[1] Similarity-Aware Token Pruning: Your VLM but Faster

[2] HeatViT: Hardware-Efficient Adaptive Token Pruning for Vision Transformers

**Questions:**

What's the main contributions from the methodology perspective?

Why not cite the ANN transformer sparsification papers which this paper's main techniques originate from? Similarity-based pruning has already existed, even the token-level sparsification has already been researched for years. But no citation is found in this paper.

---

> ### Author Response · Authors · 2025-11-20
> **Responses to Reviewer TA1d's W0**
>
> Dear reviewer TA1d, thank you for your professional and insightful comments on our paper. Your feedback is extremely valuable and plays a crucial role in improving the quality of our work. In the following, we address your questions one by one. We sincerely hope this response helps resolve your concerns.
>
> ## W0: The substantive contributions of this paper significantly overlap with previous work on ANN transformer sparsification, demonstrating limited novelty. It essentially replicates the success of existing ANN transformer sparsification approaches, with even the narrative framework bearing striking resemblances.
>
> We fully understand your concerns about our novelty. This paper focuses on token compression for Spiking Transformers, and its starting point is similar to token sparsification in ANNs: *designing different methods to retain more representative tokens or remove redundant ones, thereby reducing computation and memory by performing inference with fewer tokens*. Thus, the motivation and overall narrative resemble those of token pruning methods in ANNs. **However, TP-Spikformer is not a simple transfer of ANN token sparsification techniques to SNNs; it is specifically tailored to Spiking Transformers**. Below, we explain our two contributions in detail.
>
> *First, the heuristic IRToP scoring criteria.* Unlike ANNs, SNNs propagate information in both spatial and temporal domains. Therefore, **a key challenge for token pruning in SNNs is how to effectively exploit both spatial and temporal information without disrupting their asynchronous spiking dynamics**. We take inspiration from the human visual system that prioritizes salient regions, and design a heuristic scorer that jointly incorporates both spatial and temporal information.
> - *Spatially:* In the human visual system, spatial locations compete for saliency, and only those that quite differ from their local surroundings to persist for further processing. Correspondingly, on the feature map, we measure the representation difference between each token and its spatial neighbors, treat tokens with larger local contrast as more “spatially salient,” and assign them higher keep scores.
>
> - *Temporally:* The human visual system is highly sensitive to sudden and prominent temporal changes, which enables the brain to quickly locate and process key temporal dynamics in continuous visual input. Following this, we measure the temporal variation of each token across consecutive time steps and preferentially retain tokens that exhibit strong temporal changes.
>
> *Second, the block-level early stopping strategy.* In Appendix F, **we analyze the differences between Spiking Transformers and ANN Transformers. The key distinction is that the former interleave Transformer blocks with convolutional layers**. Specifically, early SNN transformers, such as Spikformer and SDT-V1, follow ViT-style designs from ANNs, using patch embedding and standard transformer blocks. As the field developed, recent SOTA models like QKFormer and SDT-V2/V3 incorporate convolution layers inside transformer blocks. **Unlike ANNs where features can be flattened for token removal, these convolutional layers embedded in the transformer blocks present new challenges for token compression in SNNs,** detailed as:
> - On the one hand, convolution operations rely on structured grid-like inputs. This means that **if tokens are removed from a transformer block, the remaining tokens may no longer form a valid image layout, making them incompatible with later convolutional layers**.
> - On the other hand, due to its strong prior assumptions, e.g., spatial local correlation and translation invariance, the convolution operation heavily relies on the spatial structure of feature maps. **However, removing tokens disrupts the spatial structure of feature maps. This disruption (1) impairs local information propagation, (2) degrades the effectiveness of trained filters, and (3) compromises the model's representational ability.**
>
> Thus, when performing token pruning on SOTA Spiking Transformers such as QKFormer and SDT-V2/V3, it is crucial to preserve the completeness of the feature maps and their spatial structure. This is an additional challenge that ANN-based token pruning do not need to address. Although the block-level early stopping strategy is simple, it offers a straightforward and effective way to solve this issue.
>
> It is worth noting that existing SNN token pruning methods have only been validated on relatively simple ViT-style architectures, such as Spikformer and Spike-Driven Transformer V1, and have not been extended to the aforementioned SOTA Spiking Transformers. As a result, they do not fundamentally address these additional challenges. To the best of our knowledge, we are the first to systematically analyze the unique difficulties of token pruning in Spiking Transformers and to propose a simple and general method to address this problem.

---

> ### Author Response · Authors · 2025-11-20
> **Responses to Reviewer TA1d's W1**
>
> ## W1: The spatial token scorer is highly similar to the ANN token-pruning one [1]. Thus, the novelty of this paper is significantly challenged.
>
> Thank you for your question. **Our spatial scorer is methodologically different from SAINT [1]; the only commonality is that both methods use similarity as a feature metric. However, similarity is a standard tool for measuring the closeness of high-dimensional feature vectors.** Below, we separately describe how SAINT and our scorer perform token pruning.
> - SAINT **treats the feature map as a 1D sequence of tokens and partitions all tokens into two disjoint sets** ($src$ and $dst$). For each token in src, it computes its similarity to all tokens in dst, thus **constructing a weighted bipartite graph between the two sets based on these similarities**. SAINT then applies a threshold to **keep only high-similarity edges, yielding a subgraph**. Finally, it **uses the node degree in this subgraph as a redundancy measure**: tokens in src with higher degrees, i.e., those connected to many similar neighbors, are regarded as redundant and are pruned during inference, so that more informative tokens are preserved overall.
> - Our spatial scorer **does not construct a graph or perform an $src$/$dst$ partition**. Instead, it **operates directly on the 2D feature map**. For each token on the feature map, we **compute its representational difference from its local spatial neighborhood (e.g., up, down, left, and right)**. Tokens that **exhibit larger local contrast relative to their neighbors are regarded as more “spatially salient”** and are treated as informative tokens to be preserved.
>
> In short, *SAINT adopts a graph-theoretic view, operates on a weighted bipartite graph over 1D token sequences, and defines redundancy via node degree to remove the most redundant tokens*. In contrast, *our method is inspired by human vision, operates directly on 2D feature maps, and measures local contrast to identify spatially salient tokens to be preferentially retained*. Additionally, **because convolution layers in Spiking Transformers strongly rely on a regular spatial grid, SAINT’s graph-based formulation cannot be applied for token pruning in SNNs.**
>
> > [1] Similarity-Aware Token Pruning: Your VLM but Faster

---

> ### Author Response · Authors · 2025-11-20
> **Responses to Reviewer TA1d's W2**
>
> ## W2: The ablation study in Table 6 is conducted on ImageNet, a static dataset with no temporal disparity, thus I cannot confirm the effectiveness of the temporal pruning scorer. The visualization in Figure 5 does not entirely prove the effectiveness of the temporal pruning scorer. Instead, it exhibits limitations in spatial perception of the spatial scorer.
>
> Yes, ImageNet is a static dataset. Below, we address your concerns from three aspects.
>
> 1. *First, although ImageNet is static, introducing temporal scoring is still necessary.* The SNN community typically evaluates the superiority of Spiking Transformers on this dataset, where the images are repeatedly input into the network over time. **Although the input image at each time step is identical, the hidden-layer feature maps differ across time due to membrane potential integration, leakage, and reset in spiking neurons. Therefore, tokens at the same spatial location exhibit different representations across time steps.** This is exactly the motivation behind our temporal scorer: when selecting tokens, we aim to exploit their variation along the temporal dimension. On QKFormer, the performance of full IRToP (81.16\%) > temporal-only scorer (79.69\%) > spatial-only scorer (58.93\%), showing that the temporal scorer is indeed necessary and effective.
>
> 2. *Second, we conduct additional ablation experiments on DVS-CIFAR10 to further address your questions.* **DVS-CIFAR10 is widely regarded as a large-scale DVS dataset with rich temporal dynamics.** Experiments are performed using the Spikformer architecture (network details are given in [1]), and all models are fine-tuned for 50 epochs. As shown in the table below, **the results follow the trend of IRToP > temporal-only scorer > spatial-only scorer. This further confirms the effectiveness of the temporal scorer.**
>
> We hope that the above analysis and additional experiments help to clarify and resolve your concerns.
>
> | Architecture | Dataset  | #Token | Scorer           | Accuracy |
> |-------------|---------|--------|------------------|----------|
> |Spikformer  | CIFAR-10| 0.72| IRToP | 79.9     |
> |Spikformer|CIFAR-10|0.72| only Temporal|78.8|
> |Spikformer|CIFAR-10|0.72| only Spatial| 77.1|
>
> [1] SPIKFORMER: WHEN SPIKING NEURAL NETWORKMEETS TRANSFORMER. ICLR 2023.
>
>
>
> 3. *Third, regarding the visualization of the scorer.* The temporal scorer assigns higher keep scores to tokens with stronger temporal variation. Due to the nonlinear dynamics of spiking neurons, it is difficult to precisely characterize the temporal information of static images encoded in hidden layers. Nevertheless, **Figure 5 shows that the temporal scorer tends to highlight key parts of the bird (e.g., claws, wings, beak)**, indicating that it can capture semantically discriminative regions. More importantly, as shown in table below, **ablation studies on both static and dynamic datasets confirm the effectiveness of the temporal scorer.** The spatial scorer assigns higher keep scores to tokens that are spatially distinctive from their surroundings. **In images, such saliency often appears along edges and textures. As shown in Figure 5, it tends to emphasize the bird’s contour, which is consistent with our local-contrast-based design in spatial scorer.**

---

> ### Author Response · Authors · 2025-11-20
> **Responses to Reviewer TA1d's W3**
>
> ## W3: It is very counterintuitive that the temporal scorer is used to prune spatial token rather than prune temporal token. Moreover, temporal pruning is applied regardless of whether the dataset is dynamic or static.
>
> Thank you for this insightful question. We address your concerns from three aspects.
>
> 1. *Why the temporal scorer is used to prune spatial tokens.* A token is essentially a spatial concept that represents a specific region in an image. Although it is defined in the spatial domain, in SNNs, **the token representation at the same spatial location varies over time due to membrane potential integration, leakage, and reset in spiking neurons.** It is therefore meaningful to exploit this temporal variation when selecting tokens. An intuitive example is that, *when we watch a video, we do not attend equally to all regions in each frame*. Instead, our attention tends to focus on two types of regions: **(i) regions in the current frame that are spatially salient relative to their surroundings**, and **(ii) regions at the same spatial location that exhibit strong changes across consecutive frames.** The latter motivates our temporal scoring module, which prunes spatial tokens based on their temporal variation.
>
> 2. *Why the temporal scorer is not used to prune temporal tokens.* **In fact, at each time step there is no explicit notion of a “temporal token.”** We understand your point to be that using temporal information to reduce the computational complexity along the time dimension of SNNs (e.g., by reducing the number of time steps), rather than in the spatial domain, may be more meaningful. We fully agree with this perspective. **Our work is orthogonal and complementary to such techniques, and it can be combined with them to achieve compression along multiple dimensions.**
>
> 3. *Why temporal pruning is applied on both dynamic and static datasets.* TP-Spikformer’s temporal scorer does not operate on the raw input frames, but on token representations in the hidden layers. Regardless of whether the dataset is static or dynamic, within an SNN, the representations of tokens at the same spatial location differ across time steps due to membrane potential integration, leakage, and reset in spiking neurons. Therefore, **even on static datasets, tokens are not “uninformative” along the temporal dimension. The temporal scorer is designed to exploit these dynamics to provide meaningful guidance for token selection.**

---

> ### Author Response · Authors · 2025-11-20
> **Responses to Reviewer TA1d's W4**
>
> ## W4: The retaining is already existent. This is also called token emerger technique. Or, retaining can be seen as a naive token emerger that can restore non-informative tokens for future use. I can easily give an example of such retaining [2].
>
> Thank you for your professional comment. The retaining in our block-level early stopping strategy is fundamentally different from the “token emerger” technique. Below, we first clarify the conceptual differences between token emerger and our retaining, and then use your referenced work [2] as a concrete example to illustrate this distinction in detail.
>
> We first clarify their differences.
> - Token emerger *aggregates multiple similar and redundant tokens into a single new token*, which is then processed by SSA and FFN. **Its goal is to reduce the number of tokens while preserving as much information as possible from the merged tokens, thereby lowering computational cost with minimal information loss.**
>
> - Our retaining strategy is designed to handle the architectural constraints of deeply coupled convolution-Transformer structures in Spiking Transformers. *After redundant tokens are identified, they are not aggregated but simply placed back into the feature map at their original spatial locations after block computation.* **Its main purpose is to preserve a complete 2D grid structure required by subsequent convolutional layers in SOTA models such as QKFormer and SDT-V2/V3.**
>
> Notably, in pure ViT-style architectures such as Spikformer or Spikingformer, where convolutional layers are absent, our method can directly drop redundant tokens without any retaining. Therefore, our retaining is fundamentally different from the token emerger.
>
> We then explain the difference between [2] and our method. The method in [2] divides tokens into informative and uninformative ones. Informative tokens are processed by SSA and FFN, while **uninformative tokens are aggregated into a single new token, which is then appended to the informative token sequence and processed. This method essentially treats the feature map as a 1D token sequence and does not preserve its 2D structure.** In contrast, uninformative tokens in TP-Spikformer are neither aggregated nor used in subsequent computation. Instead, **we place the updated informative tokens and the uninformative tokens back in their original spatial locations after block computation, so as to reconstruct a complete feature map**. This achieves computational reduction while preserving the 2D grid structure required by subsequent convolutional layers.
>
> > [2] HeatViT: Hardware-Efficient Adaptive Token Pruning for Vision Transformers

---

> ### Author Response · Authors · 2025-11-20
> **Responses to Reviewer TA1d's W5**
>
> ## W5: In short, this paper shows severe duplication of ANN token-pruning domain. In essense, the success of this paper's method is inherited from the ANN token-pruning works. Such that this paper's contribution is limited. Moreover, the spatial token pruning may be even better if the most advanced ANN token pruning is applied.
>
> 1. *Contribution.* As discussed above, we do not simply reuse ANN methods; instead, we design our approach around the temporal dynamics and architectural properties of Spiking Transformers. **The heuristic scorer is proposed to meet the requirement of SNNs for joint spatial–temporal modeling.** Our overlap with ANN works lies only in using similarity as a feature metric, which is merely a basic measurement tool *rather than a methodological inheritance*. **The block-level retention mechanism, although simple, is specifically designed to handle the architectural constraints of the deeply coupled convolution–Transformer structures in Spiking Transformers.**
>
> 2. *Use of the most advanced ANN token pruning method.* Regarding “spatial token pruning may be even better if the most advanced ANN token pruning is applied,” this is not necessarily the case. Token pruning in ANNs is indeed mature, and many methods achieve nearly lossless performance in training-free manners. **This is largely because mainstream ANN Transformers lack convolution layers, so feature maps can be treated as 1D token sequences, which makes token pruning much easier.** In SNNs, the situation is quite different and much more challenging: **SOTA Spiking Transformers usually include convolution layers that are highly sensitive to token pruning, and dropping or reordering tokens in intermediate layers can disrupt the spatial priors and local structures on which convolutions rely.** As a result, many advanced pruning methods in ANNs **cannot be directly applied** to Spiking Transformers, and even if extended, their performance **would likely be worse** than in ANNs due to convolution layers in SNNs.

---

> ### Author Response · Authors · 2025-11-20
> **Responses to Reviewer TA1d's Q1**
>
> ## Q1: What's the main contributions from the methodology perspective?
>
> From a methodological perspective, this paper has two contributions.
>
> - First, we propose IRToP, a heuristic spatial–temporal scoring criterion tailored to SNNs. Inspired by the human visual system, IRToP jointly considers local spatial contrast and temporal variation, providing guidance for token pruning without breaking the asynchronous nature of SNNs. Extensive ablations on both static/dynamic datasets and across multiple architectures have validated the effectiveness of IRToP.
>
> - Second, we introduce a block-level early stopping strategy for convolution–Transformer hybrid Spiking Transformers. Although its retention mechanism is not new, we point out an additional structural challenge of token pruning in SNNs and address it in a simple and general way.
>
> Beyond methodology, TP-Spikformer also makes several meaningful contributions to the SNN community:
>
> - TP-Spikformer is the first general token dropping method in the SNN domain, and it can be readily applied to a variety of architectures and tasks. It performs well across various spiking transformers without requiring modifications to network architectures. In contrast, existing SNN token drop methods either alter the backbone connection or introduce additional parameters.
>
> - TP-Spikformer provides a broader evaluation, establishing a new comparative benchmark that is currently lacking in SNN token pruning research. We validate TP-Spikformer across five tasks (image classification, object detection, semantic segmentation, event-based object tracking, and NLP) and four representative spiking transformers (simple Spikformer and SDT-V1 as well as SOTA architectures QKFormer and SDT-V3). In contrast, existing SNN token drop methods are typically limited to 1–2 simple architectures and focus solely on the image classification task.

---

> ### Author Response · Authors · 2025-11-20
> **Responses to Reviewer TA1d's Q2**
>
> ## Q2: Why not cite the ANN transformer sparsification papers which this paper's main techniques originate from? Similarity-based pruning has already existed, even the token-level sparsification has already been researched for years. But no citation is found in this paper.
>
> We apologize for the incomplete citations and discussion. Although IRToP uses similarity, this paper does not treat “similarity” itself as the core novelty. Its contribution lies in a new heuristic scoring mechanism that jointly considers local spatial contrast and temporal variation across time steps, specifically designed for SNNs. **As per your suggestion, we have now added a discussion of relevant ANN works below, such as similarity-based token selection and token-level sparsification.**
> We will include this content in the revised manuscript, thereby providing a systematic review and comparison between token compression methods in ANNs and our approach.
>
> ```
> 2.1 Token Compression in ANNs
>
> In the field of ANNs, token compression has been extensively studied. Early approaches typically introduce auxiliary modules to estimate token importance and discard redundant tokens during inference. For example, DynamicViT [1] trains an additional importance predictor to generate pruning masks and dynamically remove uninformative tokens. A-ViT [2] builds upon Adaptive Computation Time (ACT) and uses auxiliary losses to predict halting probabilities from feature channels, enabling adaptive computation. EViT [3] exploits the attention scores between each token and the class token in SSA to divide tokens into attentive and inattentive groups. The inattentive tokens are then aggregated into a new proxy token, which is concatenated with attentive tokens and fed into the following FFN, thereby mitigating the information loss caused by directly discarding them. The work [*2] you mentioned is essentially an extension of the EViT approach. However, these methods generally introduce extra parameters or rely on fine-tuning.
>
> To address this, a series of training-free token compression schemes has recently emerged as attractive alternatives. A representative example is Token Merging (ToMe) [4], which treats tokens as a 1D sequence, partitions them into two sets, constructs a weighted graph by computing pairwise similarities between tokens across the two sets, and then merges similar tokens using a bipartite matching algorithm. Subsequent works, such as ToFu [5], PPT [6], and Zero-TPrune [7], are extensions of this approach. The SAINT you referenced in [*1] is also this family's extension in the VLM domain.
>
> Despite the systematic study of token compression in ANNs, these methods are difficult to directly transfer to Spiking Transformers, mainly for the following reasons:
> (1) Most approaches flatten feature maps into 1D token sequences, whereas spiking Transformers with tightly coupled convolution–Transformer architectures must preserve a regular 2D feature grid and its spatial structure to remain compatible with subsequent spiking convolution layers that rely on local spatial priors.
> (2) Many methods depend on a cls token, while several SOTA Transformer architectures (e.g., QKFormer, SDT-V2/V3) no longer employ the cls token, making such strategies hard to apply in Spiking Transformers.
>
> Refs:
> [1] Dynamicvit: Efficient vision transformers with dynamic token sparsification. NeurIPS 2021.
> [2] A-vit: Adaptive tokens for efficient vision transformer. CVPR 2022.
> [3] Not all patches are what you need: Expediting vision transformers via token reorganizations. ICLR 2022.
> [4] TOKEN MERGING: YOUR VIT BUT FASTER. ICLR 2023
> [5] Token Fusion: Bridging the Gap between Token Pruning and Token Merging. WACV 2024.
> [6] PPT: Token Pruning and Pooling for Efficient Vision Transformers.
> [7] Zero-TPrune: Zero-shot token pruning through leveraging of the attention graph in pre-trained transformers. CVPR 2024.
> [*1] Similarity-Aware Token Pruning: Your VLM but Faster
> [*2] HeatViT: Hardware-Efficient Adaptive Token Pruning for Vision Transformers.
> ```

---

> ### Author Response · Authors · 2025-11-27
>
> Dear Reviewer TA1d,
>
> We sincerely appreciate your time and effort in reviewing our manuscript and offering valuable suggestions. As the author-reviewer discussion phase is drawing to a close, we would like to confirm whether our responses have effectively addressed your concerns. We have provided detailed responses to your concerns, and we hope they have adequately addressed your issues. If you require further clarification or have any additional concerns, please do not hesitate to contact us. We are more than willing to continue our communication with you.
>
> Best regards.

---

### Author Response · Authors · 2025-12-03
**Summary of rebuttals for AC**

## Issues and Clarifications

**Reviewer TA1d**
1. The main concern is that this work is similar to similarity-based token sparsification methods in ANNs, questioning its originality.
   - We clarify that TP-Spikformer is **tailored for Spiking Transformers**, addressing spatial–temporal dynamics and architectural constraints **that ANN methods do not face**. Moreover, the similarity is **merely a basic measurement tool**.

2. Reviewer questioned the necessity and effectiveness of the temporal scorer, especially in the static datasets.
   - We clarify that **hidden-layer tokens vary over time even for static inputs** due to SNN dynamics. **Supplemented ablations on DVS-CIFAR10 confirm the temporal scorer’s performance gain**.

3. Reviewer found it counterintuitive to use temporal scoring to prune spatial tokens.
   - We explain that **tokens are spatial entities whose representations vary over time in SNNs, making temporal variation informative for pruning.**

4. Reviewer compared our block-level retaining to token emerger techniques, questioning novelty.
   - We clarify that **our retaining retain the 2D feature map structure required for convolution–Transformer hybrids**, whereas **token emerger aggregates tokens into new ones without preserving spatial layout**.

5. Reviewer suggested applying advanced ANN token pruning for better results.
   - We argue that **ANN pruning methods cannot directly apply to SNNs due to convolution layers in Spiking Transformers**. Advanced ANN methods may fail or degrade performance in Spiking Transformers, making our tailored approach necessary.

----

**Reviewer r4rD**
1. Reviewer requested stronger justification for the spatial and temporal scorers via visualization.
   - We **provide visualizations** showing spatial scorer highlights edges/textures and temporal scorer emphasizes semantically important regions.

2. Reviewer noted that TP-Spikformer’s training-free claim seems inconsistent, citing accuracy drops on QKFormer and SDT-V3.
   - We clarify that **training-free is comparative**: TP-Spikformer avoids extra trainable modules and outperforms prior SNN pruning without fine-tuning.

3. Reviewer asked if the method can train spiking transformers from scratch.
   - **Supplemented experiments show that TP-Spikformer can be applied during training**, achieving near-original accuracy on SDT-V3 with reduced training time and GPU memory.

4. Reviewer asked about generalization to non-grid modalities.
   - We **supplement evaluation on language tasks (SpikeLM on GLUE)**, confirming its applicability beyond vision.

5. Reviewer asked how well key information is preserved during pruning and differences vs. ANN pruning.
   - We clarify that IRToP **prioritizes high-information tokens, maximizing feature extraction under constrained resources**. Unlike ANNs, spiking transformers interleave convolutions, **requiring careful token pruning to maintain spatial structure**, a challenge our method mainly addresses.

----

**Reviewer G33Y**

1. Reviewer questioned the motivation for pruning small-/medium-scale SNNs.
   * We clarify that intermediate feature maps in Spiking Transformers must be stored off-chip, and **reducing tokens significantly decreases off-chip memory access and lowers latency and energy**. So, **token pruning for SNNs is meaningful regardless of model scale**.

2. Reviewer noted the lack of discussion on overall training costs.
   * We conducted **training experiments on ImageNet**, demonstrating **reduced time and GPU memory during training**.

3. Reviewer asked about spike-driven characteristics and energy consumption of IRToP.
   * We **added discussion on spike-driven design and energy** of IRToP.

4. Reviewer questioned the extra cost of IRToP computations.
   * We provide detailed calculations for IRToP computations, confirming practical efficiency.

----

**Reviewer BtDR**:

1. Reviewer raised concerns about manually set or grid-searched pruning ratios.
   * We clarify that **grid search does not affect inference**. We also explore an **input-aware adaptive pruning scheme**.

2. Reviewer suggested pruning Attention and MLP separately.
   * We added **module-specific pruning experiments**, concluding that separate pruning **does not significantly improve performance**.

3. Reviewer suggested comparing to token merging.
   * We performed **comparisons with ToMe**, confirming the **superiority of TP-Spikformer**.

4. Reviewer asked about applicability to ANN-to-SNN conversion.
   * We applied TP-Spikformer to **conversion-based SNNs**, showing that it **generalizes well across training paradigms**.

----

During the rebuttal, Reviewer r4rD acknowledged our explanations and increased their score to 8. For the other reviewers, we addressed their concerns through **supplementary experiments and additional clarifications**. We believe our rebuttal provides **clear and convincing evidence**, effectively resolving the reviewers’ questions.

---

### Author Response · Authors · 2025-12-03
**Summary of contributions and strengths for AC**

Dear ACs,

Thank you sincerely for taking the time and effort to handle our submission. We also greatly appreciate all reviewers' constructive feedback, which is invaluable in helping us improve the paper. Here, we provide here a concise summary to facilitate the AC’s assessment.

## Contributions

TP-Spikformer makes several meaningful contributions to the SNN community:

- TP-Spikformer is the **first general token dropping method in the SNN domain**, and it can be readily applied to a variety of architectures and tasks. It performs well across various spiking transformers **without requiring modifications to network architectures**. *In contrast, existing SNN token drop methods either alter the backbone connection or introduce additional parameters.*

- TP-Spikformer provides a **broader evaluation, establishing a new comparative benchmark that is currently lacking in SNN token pruning research**. We validate TP-Spikformer across **five tasks** (image classification, object detection, semantic segmentation, event-based object tracking, and NLP) and **four representative spiking transformers** (simple Spikformer and SDT-V1 as well as SOTA architectures QKFormer and SDT-V3). *In contrast, existing SNN token drop methods are typically limited to 1–2 simple architectures and focus solely on the image classification task.*

----

## Strengths

All reviewers consistently acknowledge TP-Spikformer’s simplicity, generality, and efficiency, particularly endorsing its extensive and solid experimental evaluation.

- **Reviewer TA1d**: Highlights that token sparsification effectively boosts efficiency, supported by solid experiments across multiple tasks and datasets.

- **Reviewer r4rD**: Emphasizes the strong originality and significance of our TP-Spikformer contrasts with prior costly token pruning methods; praises the bio-inspired IRToP criterion, IR-Arc architecture, and extensive validation across diverse architectures and tasks.

- **Reviewer G33Y**: Confirms the paper is well-written and recognizes the broad experimental coverage, i.e., classification, segmentation, detection, and tracking.

- **Reviewer BtDR**: Appreciates the method’s versatility and practicality for both shallow and deep spiking transformers; training-free design is a key advantage, validated by thorough ablation studies and large-scale results.

---

### Meta-Review · Area_Chair_Pp1k · 2026-01-07

**Summary:**

This paper proposes TP-Spikformer, a training-free, general, and efficient token pruning method for spiking Transformers. Reviewers primarily questioned the paper’s originality, noting its strong similarity to existing ANN token pruning methods, and raised concerns about the validity of temporal scoring, practical motivation, and lack of adaptive pruning. However, the authors effectively addressed these through additional experiments on dynamic vision datasets, NLP tasks, and comparisons with ToMe, demonstrating the method’s training-free efficiency, architectural generality, and cross-task effectiveness.

**Reviewer Concerns:**

The reviewers' common concerns lie in the originality of the method, the validity of the temporal scoring mechanism, the practical motivation for pruning in current SNNs, the lack of comparison with alternative techniques, and limited task, model coverage.

I believe the authors have now provided an effective and comprehensive improvement to the paper. They have addressed the major concerns by adding ablation studies on DVS-CIFAR10 to validate temporal scoring, extending experiments to NLP and conversion-based SNNs, comparing with ToMe, providing visualizations of spatial and temporal scorers, and reporting reduced training time and GPU memory usage.

**Reviewer Scores:**

I think Reviewer r4rD would likely raise his score after seeing the authors’ responses.

---

### Decision · Program_Chairs · 2026-01-26

Accept (Poster)